# Vesicular transport mediates the uptake of cytoplasmic proteins into mitochondria in *Drosophila melanogaster*

Po-Lin Chen[1,2,3], Kai-Ting Huang[2,4], Chu-Ya Cheng[2], Jian-Chiuan Li[1,2], Hsiao-Yen Chan[2,4], Tzu-Yang Lin [5], Matthew P. Su [6], Wei-Yuan Yang[7], Henry C. Chang[8], Horng-Dar Wang [4] & Chun-Hong Chen [1,2,9,3✉]

Mitochondrial aging, which results in mitochondrial dysfunction, is strongly linked to many age-related diseases. Aging is associated with mitochondrial enlargement and transport of cytosolic proteins into mitochondria. The underlying homeostatic mechanisms that regulate mitochondrial morphology and function, and their breakdown during aging, remain unclear. Here, we identify a mitochondrial protein trafficking pathway in *Drosophila melanogaster* involving the mitochondria-associated protein Dosmit. Dosmit induces mitochondrial enlargement and the formation of double-membraned vesicles containing cytosolic protein within mitochondria. The rate of vesicle formation increases with age. Vesicles originate from the outer mitochondrial membrane as observed by tracking Tom20 localization, and the process is mediated by the mitochondria-associated Rab32 protein. Dosmit expression level is closely linked to the rate of ubiquitinated protein aggregation, which are themselves associated with age-related diseases. The mitochondrial protein trafficking route mediated by Dosmit offers a promising target for future age-related mitochondrial disease therapies.

[1] National Institute of Infectious Diseases and Vaccinology, National Health Research Institutes, Zhunan, Taiwan. [2] Institute of Molecular and Genomic Medicine, National Health Research Institutes, Zhunan, Taiwan. [3] Institutes of Molecular and Cellular Biology, National Taiwan University, Taipei, Taiwan. [4] Institute of Biotechnology, National Tsing Hua University, Hsinchu, Taiwan. [5] Institute of Cellular and Organismic Biology, Academia Sinica, Taipei, Taiwan. [6] Department of Biological Science, Nagoya University, Nagoya, Japan. [7] Institute of Biological Chemistry, Academia Sinica, Taipei, Taiwan. [8] Department of Biological Sciences, Purdue University, West Lafayette, IN 47907, USA. [9] National Mosquito-Borne Diseases Control Research Center, National Health Research Institutes, Zhunan, Taiwan. ✉email: chunhong@gmail.com

Mitochondrial dysfunction is a key hallmark of aging and has been linked to numerous aging-related pathologies[1]. While the accumulation of dysfunctional mitochondria has been implicated in aging, the changes that occur in mitochondrial dynamics throughout the aging process are not yet fully understood[2]. Interventions targeting improvements in mitochondrial homeostasis in elderly model organisms could, therefore, have a significant impact on lifespan and general health. However, identifying the most relevant changes in mitochondrial proteins in the context of aging remains challenging.

Mitochondrial quality control is orchestrated via mitophagy, which occurs in parallel with morphology regulation, via mitochondrial fusion and fission[3]. Mitochondrial fusion and fission processes are mediated by guanosine triphosphatases (GTPases)[4]: mitofusin (Mfn) and dynamin-related protein 1 (Drp1)[5] is required for mitochondrial fusion/fission. However, the interplay between mitochondrial dynamics and mitophagy during aging remains poorly understood.

One clear effect of aging is that aged muscles exhibit significantly enlarged mitochondria (megamitochondria)[6]. It has been shown that mouse cardiac muscle and fly flight-muscle tend to accrue enlarged mitochondria with aging[7,8]. Aged myocyte mitochondria become substantially enlarged and are thus often called "giant" mitochondria[9,10]. Aging- or disease-related mitochondrial enlargement has previously been shown to be partly due to the mitochondrial dynamic proteins optic atrophy type 1 (Opa1) or Drp1[6,11], although the detailed mechanisms have not yet been investigated.

The membrane translocase complex, TIM/TOM[12], which plays a significant role in mitochondrial protein transport, could also be involved in aging[13]. However, the existence of alternative transport pathways for delivering cytosolic proteins into mitochondria has also been suggested[14–17]. The canonical mitochondria targeting sequence (MTS) is known to control the import of proteins into mitochondria[18], but the number of mitochondrial proteins it controls is severely restricted. The processes regulating the accumulation of lysosome-like organelles within mitochondria[15], the import of misfolded proteins into mitochondria[16], and the roles of Ago2 and RISC in the translation of mitochondria-encoded genes[17] are further examples of processes in which proteins or protein complexes enter mitochondria via unknown mechanisms.

Here, we investigate a mitochondrial dynamic protein, Dosmit (downsizing mitochondria), which is implicated in mediating mitochondrial enlargement via the formatting of intramitochondrial vesicles for the uptake of cytosolic proteins into the organelle. We confirm that mitochondria become enlarged during aging, and that this is controlled by Dosmit. We find that the enlarged mitochondria contain double-membraned vesicles, and that ectopic expression of Dosmit induces a similar phenotype. Cytosolic proteins can be taken up by these intramitochondrial vesicles, which may originate from the outer mitochondrial membrane. Finally, the delivery of proteins into mitochondria via these double-membraned vesicles is coordinated by the dynamic protein and the Rab GTPase Rab32. This process appears to be associated with aging, since Dosmit knockdown results in an increased lifespan, while Dosmit overexpression results in a reduced life expectancy. We thus suggest that this is a pathological route responsible for aging and mitochondrial dysfunction.

## Results

### Aged mitochondria increase in size and contain intramitochondrial vesicles.
Skeletal muscle has a primary role in locomotion and the maintenance of posture, but it also maintains metabolic homeostasis by providing energy via the uptake of glucose and the oxidization of fatty acids[19]. For this purpose, muscle tissue contains a large number of mitochondria[20]. Building on initial qualitative imaging results revealing highly reticular mitochondrial networks in rodent muscles[21,22] and on live-cell-imaging approaches used to quantify mitochondrial morphology[23], a quantitative method has been developed that allows the quantification of mitochondrial size and shape in two dimensions[24].

We investigated mitochondrial size and shape in wild-type *Drosophila melanogaster* aged 3 days, 2 weeks, 4 weeks, and 8 weeks. We found a general increase in mitochondrial size within muscles as the flies aged (Fig. 1a, b). We then performed transmission electron microscopy (TEM) to examine the mitochondria from each age group. We found that vesicle-like structures were identifiable in aged-muscle mitochondria, but not in those in the younger muscles (Fig. 1c, d): ~20% of the mitochondria in 8-week-old flies contained intramitochondrial vesicles, whereas almost none of the mitochondria of 1-week-old flies contained such structures (Fig. 1c, d). These unusual vesicle-like structures thus serve as a marker for aging. Also, changes in mitochondrial shape, such as enlargement, that occur before signaling have been reported to influence various physiological conditions[25–30].

### Increase in mitochondrial size is not associated with the mitochondrial dynamic proteins Marf, Opa1, Drp1, and Fis1.
We investigated whether the fusion–fission machinery and the quality control mechanisms were responsible for the control of mitochondrial size during aging. Mitochondrial dynamics are known to be regulated by a set of mitochondrial proteins that includes Mitochondrial assembly regulatory factor (Marf) and Opa1[31] for mitochondrial fusion and Drp1 and Mitochondrial fission 1 protein (Fis1)[4,31] for mitochondrial fission. In order to investigate the dynamic changes in these proteins during aging, we generated HA-tagged CRISPR knock-in lines for the genes for each these proteins (Supplementary Fig. 1a–c). Marf and Opa1 HA-tagged lines indicated that the levels of these proteins were lower in aged (8-week-old) muscles than in young (3-day-old) ones (Fig. 2a, b). On the other hand, levels of the mitochondrial fission proteins Drp1 and Fis1 were higher in elderly flies (Fig. 2c, d). Unlike Marf and Drp1, which have only one major spliced form, Opa1 and Fis1 contain multiple spliced forms[32]. Since this included alternative spliced forms, aging might also cause the regulation of these dynamic genes via RNA splicing.

In single-muscle proteomic results from human patients, expression of the mitochondrial fusion protein mitofusin 2 (MFN2) and the dynamin-like protein Opa1 declines in aged muscles[19], which is consistent with our findings for flies. Mitochondrial fission also exhibits age-dependent changes. For example, the expression of the mitochondrial fission machinery, Dynamin 2 (DNM2), clearly increases with aging[19]. These observations and those of our fly aging-muscle model show a similar trend: a decrease in fusion and an increase in fission with aging.

### Dosmit is a mitochondrial dynamic protein.
The above results suggest that the aging-related enlargement of mitochondria may not be directly associated with known dynamic protein changes during aging. We therefore searched for potential genes involved in this process and performed small-scale RNAi screening focused on mitochondria-associated proteins, particularly those of the outer membrane (Supplementary Fig. 2). We found that knocking down gene *CG1458* (previously referred to as *Cisd2*) resulted in a reduction in mitochondrial size that remained consistent throughout the aging process (Fig. 2e, f). We thus

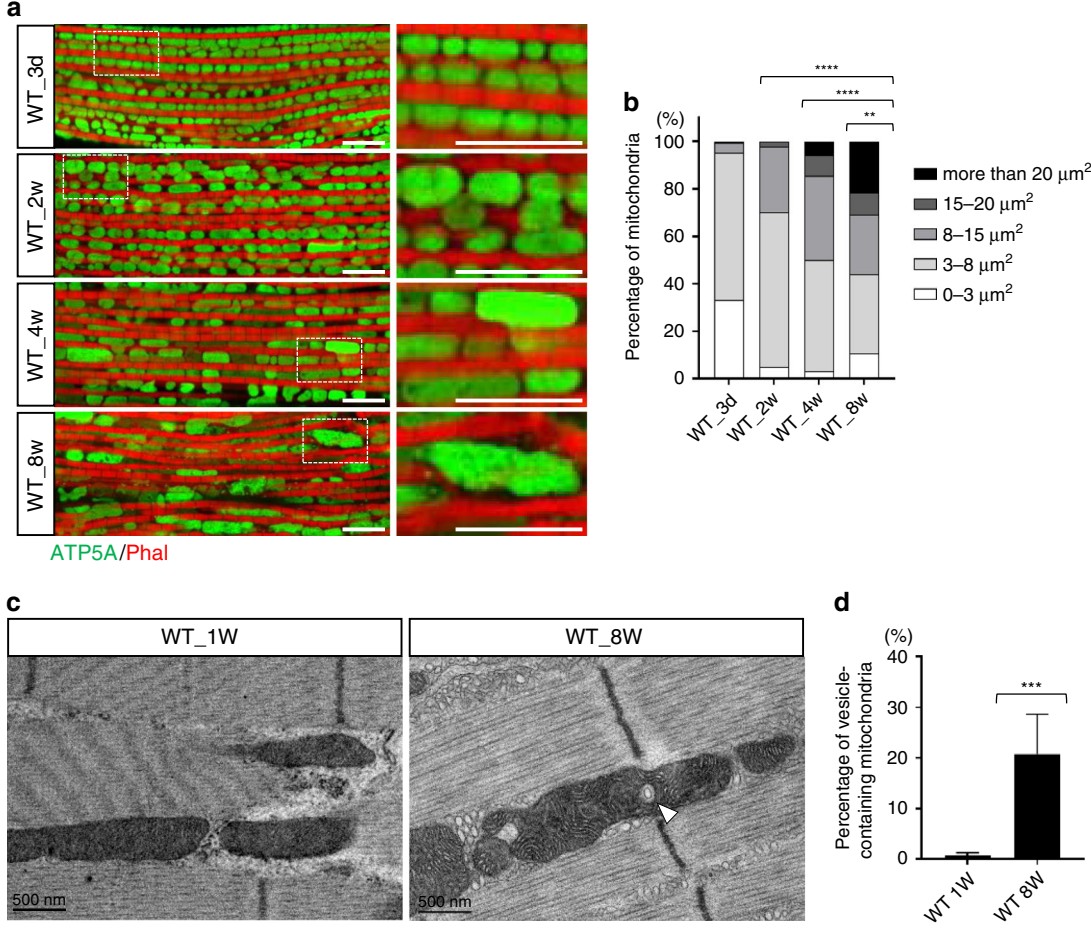

**Fig. 1 Mitochondrial size and rate of double-membraned intramitochondrial vesicle formation increases with age. a** Immunofluorescent staining of mitochondria from *D. melanogaster* aged 3 days, 2 weeks, 4 weeks, 5 eight weeks. The green channel is ATP5A staining and the red is phalloidin (Phal) staining of F-actin in the muscle. Scale bars: 10 µm. **b** Relative abundance of mitochondria of different sizes in the muscle tissue of flies of the four age groups. $N = 208, 141, 124$, and 84 from left to right bars. Statistical test: Chi-Square Test (**$p < 0.01$; ****$p < 0.0001$). **c** TEM images of intramitochondrial vesicles from flies aged 1- and 8-week-old (white arrowhead indicates an intramitochondrial vesicle). **d** The percentage of vesicle-containing mitochondria in flies aged 1 and 8 weeks (mean ± SD). $N = 718$ and 265 from left to right bars. Statistical test: Two-tailed Mann–Whitney $U$-test (***$p < 0.001$). Source data are provided as a Source Data file.

focused on the protein encoded by this gene and renamed it *Dosmit* (downsizing mitochondria).

Dosmit is an iron–sulfur-binding protein containing a C-terminal CDGSH domain defined as an iron–sulfur-binding structure (Supplementary Fig. 14a). We found Dosmit localized primarily in the mitochondria (Fig. 2g), and also colocalized with mitochondrial outer membrane protein Tom20 in adult muscle (Supplementary Fig. 3a), implying that Dosmit may be localized to the outer membrane. To determine the location of Dosmit, we used transgenic fly lines with KDEL-fused GFP as an ER marker, and a Golgi-GFP. We found most of the Dosmit signal on the edges of mitochondria (Supplementary Fig. 18). We also examined Dosmit localization in vitro by digitonin and proteinase K digestion in S2 cells, Dosmit could be degraded as Porin, suggesting that Dosmit is relatively accessible and likely located on the outer membrane of mitochondria (Supplementary Figs. 19 and 20). We showed that Dosmit existed in monomer and dimer (Dosmit*) forms, which were detected in the mitochondrial fraction of the flight-muscle (Fig. 2h). To confirm that the upper-band signal is Dosmit dimer, we analyzed the Dosmit protein in the presence and absence of the reducing agent Dithiothreitol (DTT) (Supplementary Fig. 4h). DTT was able to reduce the dimer form of Dosmit to its monomer form in wild-type flies. However, neither form was detected in the

Dosmit mutant via transposon integration (*Dosmit-EP/cisd2*[G6528]) (Supplementary Fig. 4a).

To explore the roles of endogenous Dosmit, we characterized flies that were homozygous for a Dosmit mutation (Fig. 2i). Mitochondria from *Dosmit-EP* mutants were generally more rounded similar to those in the RNAi line (Fig. 2e, j), and approximately three times smaller than those of wild-type flies (Fig. 2k). To confirm this finding, we generated another Dosmit mutant (*Dosmit-2-1*) using the CRISPR/Cas9 system (Supplementary Fig. 4b). Flies homozygous for *Dosmit-2-1* exhibited phenotypes similar to those observed in *Dosmit-EP* and RNAi flies (Supplementary Fig. 4c–f). The smaller-mitochondria phenotype of *Dosmit EP* and *Dosmit-2-1* was fully rescued by the genomic clone (Supplementary Fig. 5c, g, i, j).

These results show that Dosmit is required to maintain normal mitochondrial size and shape and that Dosmit is also involved in regulating mitochondrial dynamics.

**Dosmit expression level was associated with aging and mitochondrial size.** There is a positive correlation between muscle-tissue aging and mitochondrial size (Fig. 1a, b). To investigate the age-dependent functionality of Dosmit in terms of mitochondrial

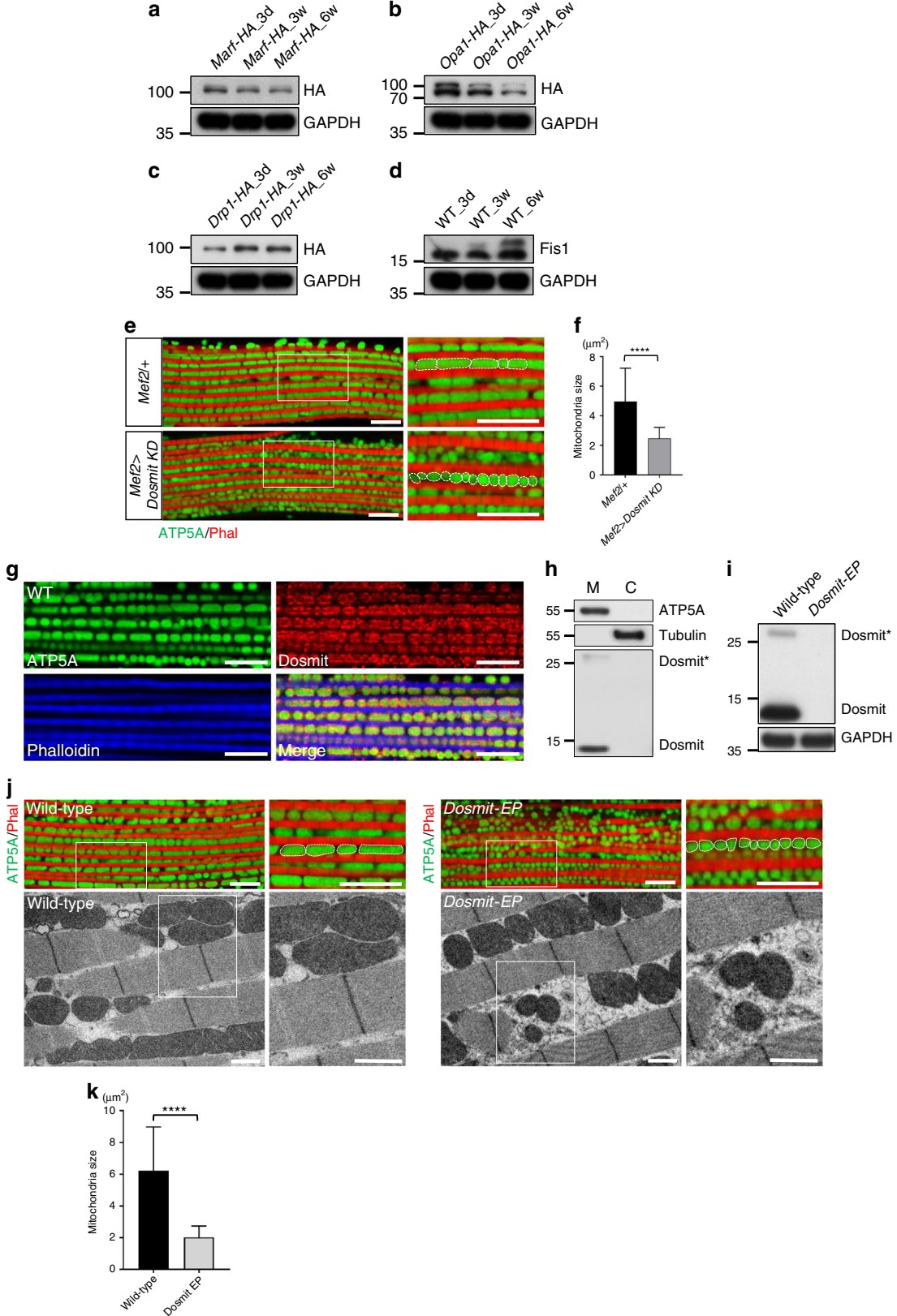

size, we examined Dosmit protein expression levels in flies of different ages.

During the aging process, Dosmit expression in wild-type flies increased with age. This was particularly true for the dimer form (Fig. 3a, b): while the monomer form increased slightly in old age, the dimer form increased by up to 500%. However, other mitochondrial proteins, Porin, ATP5A, Ndufs3, Chchd3 as control did not change significantly during aging (Supplementary Fig. 6). Aged flies, which had higher levels of Dosmit, also had larger mitochondria than younger flies (Fig. 3c, d). We investigated whether the increase in the level of Dosmit causes changes in the levels of the dynamic proteins Drp1, Marf, Opa1,

**Fig. 2 Knocking down a mitochondria-associated protein, Dosmit, results in significant differences in mitochondrial size and shape. a–d** Western blots highlighting expression levels of mitochondrial proteins: **a** Marf, **b** Opa1, **c** Drp1, and **d** Fis1. **e** Confocal microscopy of mitochondria in control and Dosmit-knockdown (KD) flies, highlighting the size and shape of mitochondria. Scale bar: 10 μm. **f** Mitochondrial size in control and Dosmit-knockdown flies (mean ± SD). $N = 126$ and 144 from left to right bars. Statistical test: Two-tailed Mann–Whitney $U$-test (****$p < 0.0001$). **g** Immunofluorescent staining of muscles of wild-type flies: ATP5A (green) in the inner membrane of mitochondria; F-actin (blue) in muscle fibers, between which oval-shaped mitochondria are aligned; and Dosmit (red) localized to mitochondria. Scale bar: 10 μm. **h** Western blotting of flight-muscle tissue homogenates indicating the presence of Dosmit protein in the mitochondrial fraction only (M, mitochondrial fraction; C, cytosolic fraction; tubulin: cytosolic marker; Dosmit*: Dosmit dimer). **i** Western blotting of flight-muscle testing the presence/absence of Dosmit protein in tissue homogenates from control and homozygous *Dosmit-EP* mutant flies (GAPDH: loading control). **j** Confocal microscopy of mitochondria in the flight muscles of control and homozygous *Dosmit-EP* flies. Color panels are confocal microscopy images; gray-scale panels are TEM images. Scale bar of fluorescence images: 10 μm; scale bar of TEM images: 1 μm. **k** Size (mean ± SD) of mitochondria in flight-muscle of wild-type and *Dosmit-EP* flies. $N = 91$ and 131 from left to right bars. Statistical test: Two-tailed Mann–Whitney $U$-test (****$p < 0.0001$). Source data are provided as a Source Data file.

and Fis1. However, knockdown of Dosmit had no effect on the levels of these proteins during the aging process (Supplementary Fig. 7). We also tracked mitochondrial size during aging in *Drp1-*, *Marf-*, and *Opa1*-heterozygous mutants, and found similar patterns of mitochondrial enlargement in all three lines (Supplementary Fig. 8). These results confirm that the increase in mitochondrial size during aging does not primarily depend on the known mitochondrial dynamic proteins Marf, Opa1, or Drp1.

The mitochondria of homozygous *Dosmit-EP* mutants were significantly smaller than those of wild-type flies of the same age. This small-mitochondria phenotype was maintained up to 8 weeks of age (Fig. 3e, f), further suggesting that changes in Dosmit expression levels during aging play a crucial role in regulating mitochondrial size. Combined, these results suggest that mitochondrial enlargement might be an aging-related process and that the level of Dosmit is at least indirectly associated with mitochondrial enlargement.

**Ectopic expression of Dosmit enlarges mitochondria and induces the formation of intramitochondrial vesicles.** To confirm that the aging-related increase in Dosmit levels results in mitochondrial enlargement, the ectopic Dosmit expression was driven by muscle-specific Mef2-GAL4. In *Mef2 > Dosmit* flies, mitochondria were up to seven times larger than in the control flies (Fig. 4a, b).

To investigate the effects of the Dosmit-induced mitochondrial enlargement on longevity, we examined the median lifespan of various strains of *D. melanogaster*. We found that male homozygous *Dosmit-EP* flies or *Dosmit-2-1* flies lived 17% or 21% longer than male wild-type flies, but that the rescue lines showed similar lifespans to wild-type flies (Supplementary Fig. 9a). Ectopic expression of Dosmit decreased the flies' lifespan by 23% relative to control flies (Supplementary Fig. 9b). Changes in mitochondrial morphology may therefore affect mitochondrial function and result in corresponding changes in lifespan. To confirm whether the longer lifespan was associated with health-related markers of aging, we assessed the locomotion activity of these Dosmit gain- or loss-of-function lines. The *Dosmit-EP* and *Dosmit-2-1* flies exhibited better locomotion in middle age than the wild-type and Dosmit-overexpressing lines (Supplementary Fig. 10). This indicates that the loss of this protein may benefit longevity as well as locomotion, suggesting that Dosmit mutants may partially suppress locomotion defects.

The enlarged mitochondrial phenotype was observed in both adults and larvae (Fig. 4a, b), TEM images showed the enlarged mitochondria were associated with the formation of numerous intramitochondrial vesicles, each of which was surrounded by a double membrane (Fig. 4c, d). These vesicles were found exclusively in aged muscles (Fig. 1c, d), suggestive of a positive correlation between a high level of Dosmit expression and vesicle formation.

To date, only two other types of double-membranes organelles are known to exist in cells: autophagosomes and mitochondria themselves. In order to determine whether the double-membraned intramitochondrial vesicles originate from autophagosomes, we used Atg8a RNAi in combination with ectopic Dosmit expression to interfere with autophagosome formation. The vesicles still existed after the knockdown (Supplementary Fig. 22), suggesting they originate via another mechanism. We found a reduction in electron dense material within these vesicles compared to wild-type mitochondria (Fig. 4c, d, arrow-heads), suggesting that these vesicles may be involved in the uptake of cytosolic content into mitochondria.

**Cytosolic proteins can be taken into intramitochondrial vesicles.** Nuclear-encoded mitochondrial proteins are believed to be imported from the cytosol into mitochondria via a canonical pathway involving mitochondrial translocases, specifically the TIM/TOM complex[12]. However, mitochondrial targeting sequences have not been identified in most nuclear-encoded mitochondrial proteins. Thus, these proteins may be entering mitochondria via a different method than the canonical tunnel. We therefore speculated that the Dosmit-induced vesicles may be involved in the uptake of cytosolic proteins. We thus tested whether the contents of these vesicles were of cytosolic origin.

In the muscles of adult wild-type, GFP was localized only in the cytoplasm of myocytes (Fig. 5a). However, when GFP was coexpressed with Dosmit, and GFP was often present in these enlarged mitochondria as numerous puncta (Fig. 5b). The three-dimensional video also showed that cytosolic GFP was located within the mitochondria (Fig. 5g and Supplementary Movie 1). This suggests that GFP was delivered into the enlarged mitochondria when Dosmit was highly expressed.

To confirm that these puncta signals were indeed GFP, we purified mitochondria from the muscle tissues of the wild-type flies and from the muscle tissues in which Dosmit was coexpressed. The cytosol and mitochondria fractions were subjected to sodium dodecyl sulfate–polyacrylamide gel electrophoresis (SDS-PAGE) analysis. GFP was detected only in the purified mitochondrial fraction came from flies coexpressing GFP and Dosmit, but not in that from wild-type flies nor in that in which GFP was ectopically expressed alone (Fig. 5c). Based on a quantification of the band intensity, we estimated that ~15% of the cytosolic GFP was internalized when Dosmit was ectopically expressed (Fig. 5c, d). In the muscles of adults in which GFP was coexpressed with Dosmit, after digitonin and proteinase K digestion, GFP was localized only in the mitochondria as numerous puncta (Supplementary Fig. 21b). Cytosolic GFP was mostly degraded, suggesting that the puncta of GFP were located within the mitochondria (Supplementary Fig. 21b). The same result from Western blot, showing GFP in the mitochondrial fraction are resistant for degradation (Supplementary Fig. 21c).

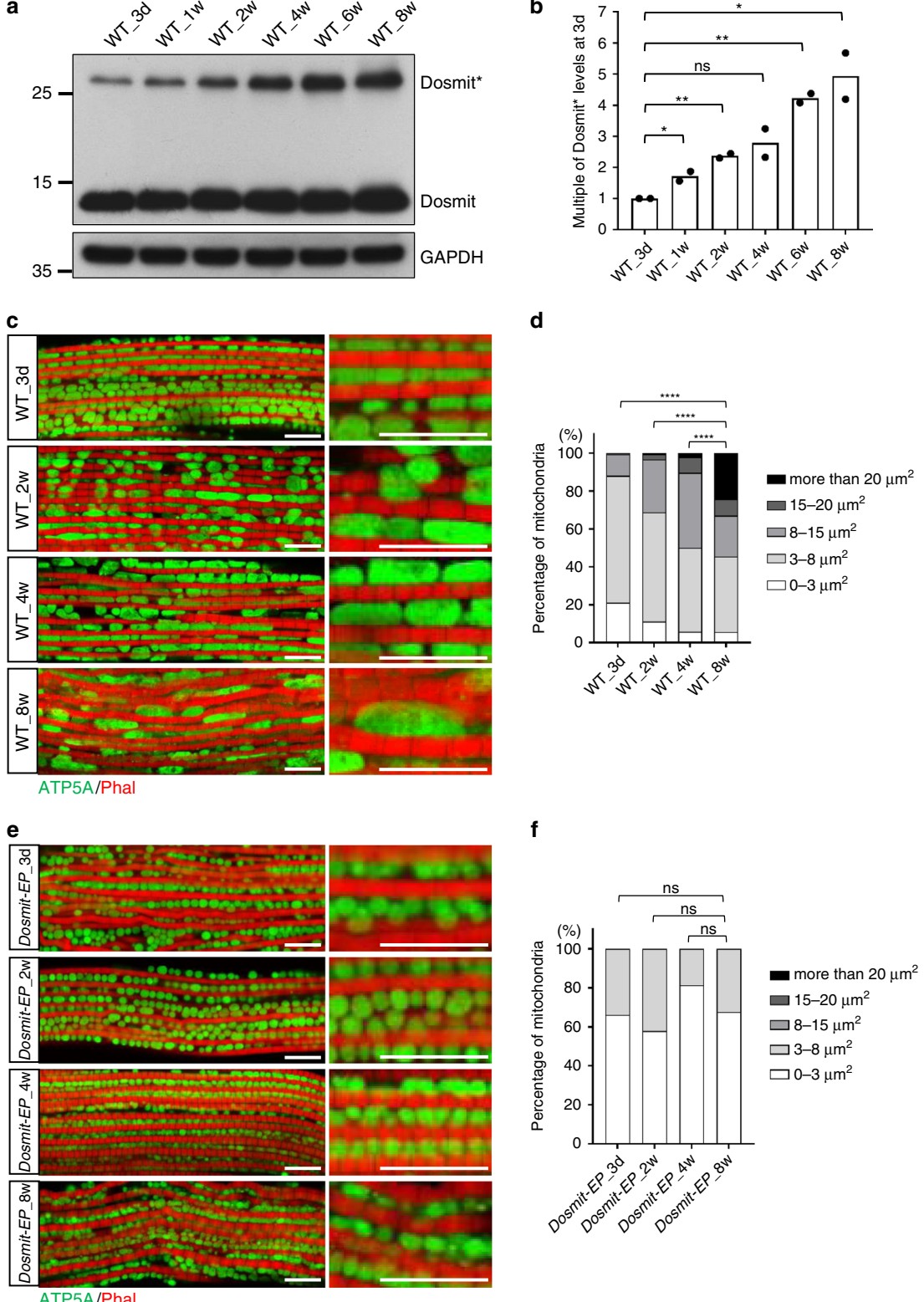

**Fig. 3 Dosmit protein expression levels increase in wild-type flies throughout natural aging. a** Western blot of Dosmit in flight-muscle tissue homogenates from wild-type flies aged 3 days, 1 week, 2 weeks, 4 weeks, 6 weeks, or 8 weeks. Dosmit* represents dimerized Dosmit. GAPDH: loading control. **b** Dosmit* expression levels based on the western blot of each age group. $N = 2$. Statistical test: two-tailed student's $t$-test (**$p < 0.01$; *$p < 0.05$; ns: $p = 0.0604$). **c** Immunofluorescent staining showing mitochondria of wild-type flies aged 3 days, 2 weeks, 4 weeks, or 8 weeks. Scale bar: 10 μm. **d** Relative abundance of mitochondria of different sizes for each wild-type fly age group. $N = 133, 144, 86$, and $106$ from left to right bars. Statistical test: Chi-Square test (****$p < 0.0001$). **e** Immunofluorescent staining showing mitochondria of *Dosmit-EP* flies aged 3 days, 2 weeks, 4 weeks, or 8 weeks. Scale bar: 10 μm. **f** Relative abundance of mitochondria of different sizes for each *Dosmit-EP* fly age group. $N = 233, 211, 192$, and $210$ from left to right bars. Statistical test: Chi-Square test (ns: $p = 0.0509, 0.1873, 0.8805$ from left to right comparison). Source data are provided as a Source Data file.

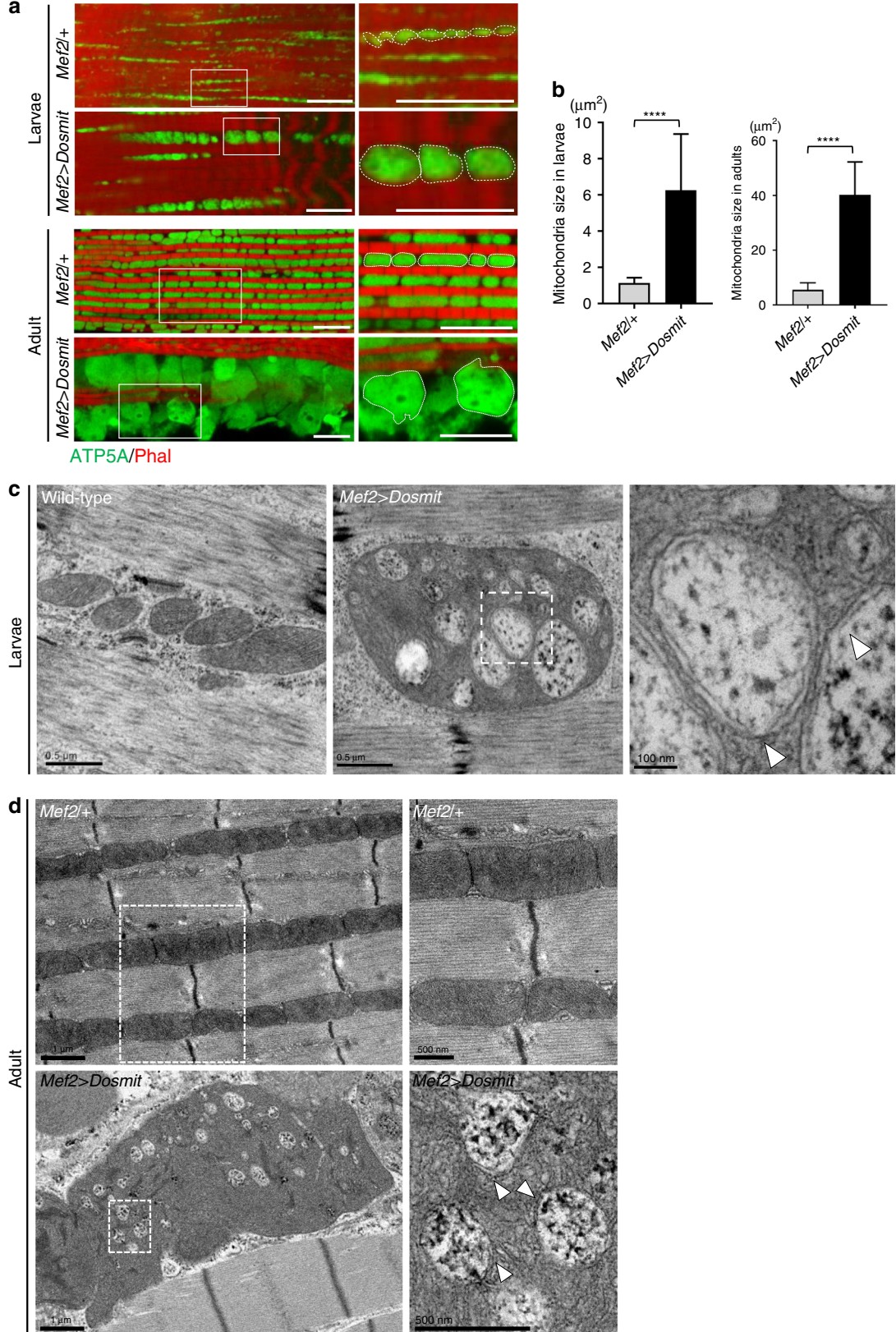

To further confirm whether the signal from the puncta in the intramitochondrial vesicles was generated by GFP, we conducted immunogold labeling of GFP (Fig. 5e). Significantly more gold particles were localized in the intramitochondrial vesicles of the enlarged mitochondria than in the mitochondria in control muscle (Fig. 5e, f). Dosmit-induced intramitochondrial vesicle formation therefore functions as part of a process controlling mitochondrial importation of cytosolic molecules.

**Intramitochondrial vesicle membranes contain the outer-membrane protein Tom20.** Previous reports indicate that

**Fig. 4 Dosmit-induced mitochondrial enlargement and formation of intramitochondrial vesicles in larvae and adult flies. a** Immunofluorescence and TEM images indicating that ectopic expression of Dosmit causes mitochondrial enlargement and the formation of double-membraned intramitochondrial vesicles in the muscles of both larvae and adult flies. Scale bar: 10 μm. **b** Mitochondrial sizes in *Mef2 > Dosmit* larvae (left) and adult flies (right) compared to *Mef2/+* controls (mean ± SD). From left to right bars, $N = 20$ and 20 (larvae); 85 and 42 (adults). Statistical test: Two-tailed Mann–Whitney *U*-test (****$p < 0.0001$). **c, d** TEM images showing the double membrane (arrowheads) of the intramitochondrial vesicles in the flight muscles of *Mef2 > Dosmit* larvae (**c**) and adult flies (**d**). Scale bars for larvae images (left to right): 0.5 μm, 0.5 μm, and 100 nm. Scale bars for adult fly images: 1 μm (left) and 500 nm (right). Source data are provided as a Source Data file.

proteins are brought to their destination in the cell by way of small vesicles that travel through the cell from an originating intracellular compartment[33,34]. Small vesicles are created by the action of coat proteins that deform membranes into vesicles and simultaneously select "cargo" proteins for inclusion in these vesicles[35,36]. Coat protein complex II (COPII) is a set of highly conserved proteins that is responsible for creating small vesicles that originate from the endoplasmic reticulum (ER)[37,38]. The formation and movement of these COPII vesicles are crucial first steps in the cellular secretion pathway through which membrane and luminal cargo proteins are transported from their site of synthesis at the ER to other membrane compartments in the cell.

Intramitochondrial vesicles may form in a similar way, from the outer membrane of the mitochondrion. The double membrane may form a concavity directed toward the inside of the mitochondrion, which would then bud off to form a double-membraned vesicle that imports cargo from the cytosol into the mitochondrion. We reasoned that such vesicles may contain outer-membrane markers, such as Tom20, which is a component of the TOM complex[12].

In wild-type flies, Tom20-HA was localized to the periphery of mitochondria (Fig. 6a). In contrast, in Dosmit-overexpressing flies, Tom20-HA was visible in circular structures within the mitochondria (Fig. 6b). Hence, in these flies, Tom20 was present not only in the mitochondrial membrane, but also surrounded the intramitochondrial vesicles.

Further immune EM labeling showed that most of the anti-HA gold particles were located on the mitochondrial outer membrane in the control (Fig. 6c, d). However, in overexpressed-Dosmit flies we observed Tom20-HA on ring-shaped structures within mitochondria (Fig. 6e–g), implying that the vesicular membrane may have come from the mitochondrial outer membrane. This provides further evidence that Tom20 occurs in the membrane of the vesicles. Altogether, these results suggest that the membranes of the Dosmit-induced intramitochondrial vesicles containing GFP are derived from the outer mitochondrial membrane (Fig. 6h).

**Aged and enlarged mitochondria import cytosolic GFP.** Given our findings thus far, we next investigated whether the presence of the vesicles and the uptake of cytosolic proteins into mitochondria were signs in aged mitochondria. GFP puncta were visible within about 20% of the mitochondria of 8-week-old flies ectopically expressing GFP (Fig. 7a, arrow-heads). In contrast, we did not observe any GFP signal within the mitochondria of 1-week-old flies, and no GFP signals observed in *Dosmit-EP* mutant and knocking down muscle of neither 1 week nor 8-week-old (Fig. 7a, b), suggesting Dosmit regulated the spontaneous appearance of these structures with aging. Ultrastructure images showed 8-week-old wild-type flies, ~12% of mitochondria contained vesicles or were undergoing the invagination process. However, no such mitochondria were observed in 1-week-old wild-type or mutant flies, nor in 8-week-old Dosmit mutants (Fig. 7c, d).

**Dosmit induces the formation of ubiquitinated proteins-containing vesicles in mitochondria.** In yeast, mitochondria have been reported to be involved in the breakdown of unfolding proteins[16], and aging in *Drosophila* has been characterized by the accumulation of protein aggregates and mitochondrial dysfunction[39]. In aged yeast, aggregations of cytosolic proteins are imported into mitochondria[16]. To further understand the content of the vesicles associated with this aging marker, we used anti-ubiquitin antibody to examine whether ubiquitinated aggregates could be transported into mitochondria via these vesicles. While ubiquitinated proteins accumulated in intramitochondrial vesicles when Dosmit was ectopically expressed and in aged wild-type flies, they did not in *Dosmit-EP* or *Dosmit-2-1* mutant flies (Supplementary Fig. 11). These results indicate that the ubiquitinated protein might be enclosed in intramitochondrial vesicles, and suggests the potential function of Dosmit on protein homeostasis.

We wanted to test the hypothesis that *Drosophila* mitochondrial proteases play a similar role in the degradation of ubiquitinated proteins within Dosmit-induced vesicles and identify the potential function of these vesicles[16]. The mitochondrial Lon protease RNAi produced hyper-ubiquitinated proteins within these vesicles, and induced severer accumulation of ubiquitinated proteins when co-overexpressing with Dosmit (Supplementary Fig. 17a–e). There were synergistic lethal phenotype when Lon RNAi and high Dosmit expression were combined (Supplementary Fig. 17f). This suggests that Dosmit and Lon are both involved in mitochondrial protein homeostasis.

**Rab32 interacts with Dosmit and locates on mitochondria that is mutually dependent.** Most membrane trafficking relies on the Rab family of small GTPases that controls vesicle fusion: they facilitate vesicle fusion by bringing vesicles into close proximity with one another. Rab GTPases play crucial roles in intracellular vesicle trafficking[40–42], and are required for precise targeting to distinct organelles[43]. Therefore, Rab GTPases provide a window into understanding how membrane trafficking is regulated. In mammals, Rab32 localizes to both the ER and mitochondria[44,45], and is involved in the regulation of ER–mitochondrion interactions and mitochondrial dynamics[46,47]. In mammalian neurons, the hyperactivation of Rab32 results in enlarged mitochondria relative to those of wild-type neurons[48].

To gain insight into the protein-trafficking mechanisms of the intramitochondrial vesicles, we examined the effect of Rab GTPases via a Rab protein screen (using Rab32 as a target). Rab32 is localized to edge of mitochondria (Fig. 8a), and purified mitochondrial fractions indicated that Rab32 occurs mainly in the mitochondria (Fig. 8d). Coexpression of Tom20-HA, YFP-Rab32 and Dosmit were all colocalized with Tom20-HA, also suggesting they likely locate on the outer membrane (Fig. 8c and Supplementary Fig. 3). Further, Rab32 did not simply share a location with Dosmit (Fig. 8b). A co-immune pull-down assay in S2 cells revealed Dosmit protein in the Rab32-HA-contained immunocomplex that was pulled down by the HA antibody

(Fig. 8e), and Rab32-HA protein but not Hsc70-5-HA in the Dosmit-associated complex that was pulled down by the Dosmit antibody (Fig. 8f). This indicates that Dosmit and Rab32 form a protein complex.

We identified a mutual dependency between Dosmit and Rab32. Dosmit is located mainly on mitochondria, likely on the outer membrane. However, in Rab32 mutants, Dosmit failed to locate on mitochondria (Fig. 9a, b). Mislocalization of YFP-Rab32

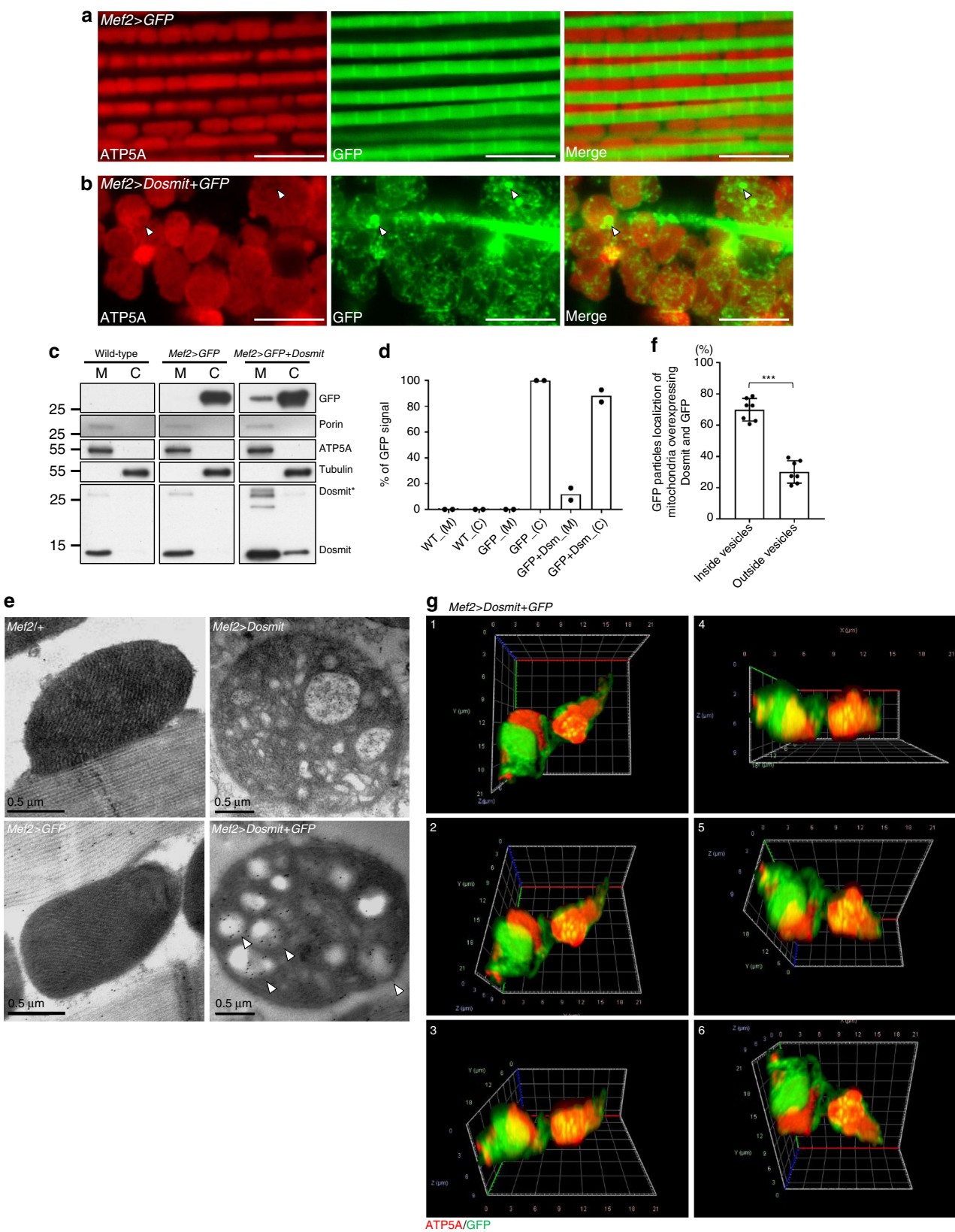

**Fig. 5 Ectopic expression of Dosmit induces the formation of intramitochondrial vesicles containing cytosolic proteins. a** Confocal microscopy showing expression of GFP in the cytoplasm of control fly muscle tissue (*Mef2 > GFP*). **b** Confocal microscopy showing localization of GFP in mitochondria in the muscle tissue of flies ectopically expressing Dosmit (arrow-heads) (*Mef2 > Dosmit + GFP*). Scale bars: 10 μm. **c** Western blot showing GFP in the mitochondrial fraction of flight-muscle homogenates from flies ectopically expressing Dosmit (*Mef2 > GFP + Dosmit*), but not in the mitochondrial fractions of flight-muscle expressing GFP without ectopic expression of Dosmit (*Mef2 > GFP*) or control flies (wild-type). **d** Quantification of band intensity showed that ~15% of the cytosolic GFP was internalized in mitochondria when Dosmit (Dsm) was ectopically expressed. $N = 2$. **e** Localization of gold-labeled GFP within the intramitochondrial vesicles of flight muscles from flies ectopically expressing Dosmit (*Mef2 > Dosmit + GFP*, arrowheads) and from mitochondria of flight muscles expressing GFP without ectopic expression of Dosmit (*Mef2 > GFP*) and control mitochondria (*Mef2/+*). Color panels are confocal microscopy images; gray-scale panels are TEM images. Scale bars: 0.5 μm. **f** GFP–gold particles localized inside and outside the intramitochondrial vesicles in muscles from flies ectopically expressing Dosmit (*Mef2 > Dosmit + GFP*). $N = 7$ and 7 from left to right bars (mean ± SD). Statistical test: Two-tailed Mann–Whitney *U*-test. (***$p < 0.001$). **g** Three-dimensional imaging of *Mef2 > Dosmit + GFP* fly mitochondria, with cytosolic GFP internalizing in the mitochondria. Source data are provided as a Source Data file.

was also detected in mitochondria of Dosmit-knockdown flies (Fig. 9c, d). The mitochondrial localization of Rab32 and Dosmit proteins is thus mutually dependent.

**Rab32 plays a critical role in regulating Dosmit-driven mitochondrial enlargement and vesicle formation**. Given the mutual dependency between Dosmit and Rab32, Rab32 is a likely candidate for involvement in Dosmit-induced mitochondrial enlargement and intramitochondrial vesicle formation. To test this, we generated both homozygous mutants, as well as flies ectopically expressing Rab32. Ectopic expression of Rab32 by itself partially induced mitochondrial enlargement, while the mitochondria in homozygous mutants were smaller (Fig. 10a–c). The Dosmit-induced phenotype was enhanced in flies coexpressing wild-type Rab32 (Rab32$^{WT}$) or in constitutively active GTP-binding Rab32 (Rab32$^{CA}$) (Fig. 10g, h, k), but was partially suppressed in flies coexpressing a dominant-negative GDP-binding Rab32 (Rab32$^{DN}$) or in Rab32-null mutant flies (Fig. 10i–k). Furthermore, mitochondria from the Rab32$^{WT}$ and Rab32$^{CA}$ flies had more intramitochondrial vesicles induced by Dosmit (Fig. 10o, p, s), but that the Rab32$^{DN}$ and Rab32 mutants had fewer (Fig. 10q, r, s).

Both Rab32 homozygous mutants and flies ectopically expressing Rab32$^{DN}$ exhibited a circular mitochondrial phenotype similar to that seen for *Dosmit-EP* flies (Figs. 2j, 10c and Supplementary Fig. 12c). In contrast, ectopic expression of Rab32$^{WT}$ or Rab32$^{CA}$ induced slight mitochondrial elongation (Fig. 10b, d and Supplementary Fig. 12b, d). These results suggest that Rab32 is involved in the upregulation of the Dosmit-induced mitochondrial enlargement effect. Taken together, our data suggests that Dosmit-induced transport of vesicles into mitochondria which is dependent upon Rab32 could provide a route for mitochondrial protein trafficking and enlargement.

**Dosmit-induced mitochondrial enlargement may be suppressed by Drp1-induced mitochondrial dynamics**. In *Caenorhabditis elegans*, an aging-related decline in mitophagy coincides with mitochondrial enlargement[49]. Drp1 is essential for mitophagy, and the induction of Drp1 at the approximate midpoint of the normal fly lifespan suppresses the occurrence of elongated mitochondria in flies that are nearing the end of their lifespan, thus prolonging their lifespan[50]. We, therefore, tested whether Drp1 could also play a role in suppressing Dosmit-induced mitochondrial enlargement.

We found that the expression of Drp1 in flight-muscle tissue partially rescued the Dosmit-induced enlarged mitochondrial phenotype (Supplementary Fig. 13f, g), implying that Dosmit-induced phenotypes might be associated with Drp1-induced mitochondrial dynamics.

**The N-terminus and CDGSH domains were required for Dosmit-induced mitochondrial enlargement**. In order to analyze which domains of Dosmit are required for mitochondrial enlargement, we divided Dosmit into multiple functional domains, including the N-terminus, transmembrane domains (TM), and CDGSH domain (Supplementary Fig. 14a). The CDGSH domain is important in that it could regulate the iron–sulfur binding required for mitochondrial oxidative capacity[51]. Fly muscle ectopically expressed Dosmit lacking CDGSH domain or N-terminus did not exhibit mitochondrial enlargement, it suggests CDGSH domain and N-terminus of Dosmit were required for the enlarged mitochondria phenotype (Supplementary Fig. 14d–f), suggesting that not only the N-terminus but also iron–sulfur binding may play an important role in regulating mitochondrial dynamics.

**Mouse Cisd-domain hybrid protein may cause mitochondrial enlargement and rescue *Drosophila* Dosmit mutants**. To test whether the Dosmit-induced mitochondrial phenotype is conserved across species, we identified and generated mouse complementary DNA (cDNA) clones for the mammalian homolog of Dosmit, Cisd. This protein family contains two members, Cisd1 and Cisd2[51,52]; Cisd2 has a longer N-terminal domain, similar to Dosmit (Supplementary Fig. 15a), but its subcellular localization is in the ER, whereas Cisd1 is located mainly on mitochondria (Supplementary Figs. 15d, e and 16a, b). Dosmit thus apparently combines the features of Cisd1 and Cisd2 (Supplementary Fig. 15a), in that it is expressed in fly muscle, S2 cells, and HeLa cells, and it localizes on mitochondria (Supplementary Fig. 16a, b).

To understand whether Dosmit and the Cisd family are functionally conserved proteins, mouse-Cisd1 (mCisd1), mouse-Cisd2 (mCisd2), or a hybrid protein that fused the long mouse-N-terminal domain of Cisd2 (1-40) with Cisd1 (11-108)/(N-mCisd2- mCisd1) ectopically expressed in *Dosmit-2-1* mutant (Supplementary Fig. 15). We found that the rounded mitochondria phenotype was only suppressed by expression of the N-mCisd2-mCisd1 hybrid protein, but not by mCisd1, mCisd2 expression (Supplementary Fig. 15f), although some of the mitochondria became larger than in wild-type cells (Supplementary Fig. 15). In addition, N-mCisd2-mCisd1 resulted in mitochondrial enlargement in both *Drosophila* S2 cells and human HeLa cells (Supplementary Fig. 16a–d). These results suggest that Dosmit is an earlier evolutionary form of the Cisd protein family.

## Discussion

The morphology of mitochondria is intrinsically linked to their function, with numerous studies finding correlations between aging, metabolic perturbations, mitochondrial morphology, mitochondrial function, and tissue function in model

organisms[6,53–56]. Here, we identified Dosmit as an essential player in aging-related enlargement and intramitochondrial vesicle formation in muscle mitochondria.

There are more recent reports on vesicle formation and trafficking from mitochondria, with mitochondria-derived vesicles (MDVs) have been reported to selectively import mitochondrial proteins and fuse with inter organelles[57]. MDVs are also important for mitochondrial quality control, dynamics, and antigen presentation[58]. Recent reports suggest that MDVs deliver the peroxide-generating enzyme Sod2 to bacteria-containing phagosomes[59]. In yeast, mitochondrial proteins are delivered to the vacuole for degradation, the vacuole being a double-membraned vesicle (MDC) itself[60]. MDCs contain both inner- and outer-mitochondrial-membrane-associated proteins, and the proteins imported into MDCs are already present within mitochondria[60]. Furthermore, the import receptor Tom70/71 required for the formation of the MDC[60]. A sorting and segregation system could therefore exist in the inner and outer mitochondrial membranes, although there is no direct evidence to show whether MDCs have a single or double membrane.

Here, we show that Dosmit may interact with the membrane-trafficking Rab protein Rab32. Rab proteins are found in all eukaryotic cells, where they mediate the fundamental processes of vesicle sorting and transport between target membranes[40]. Consequently, Rab GTPases are commonly used as markers and identifiers of various organelles and vesicles in the endocytic and secretory systems[61]. The precise effector of Rab32-downstream effectors in this process has yet to be identified. Rab32 interacts genetically with Dosmit to regulate mitochondrial enlargement and vesicle formation (Fig. 10), and both are mutually dependent for mitochondria localization (Fig. 9). We also showed that there is a direct physical interaction between the two proteins (Fig. 8e, f), although how they coordinate with each other, and the corresponding protein contributing to the altered membrane curvature, the formation of the vesicle, and the mechanisms of release or fusion with the inner membrane are also still to be identified.

In bacteria such as *Gemmata obscuriglobus*, membrane-internalization vesicles enable the uptake of external proteins[62], suggesting that mitochondria might have retained such membrane-trafficking behavior. The site of generation for Dosmit-mediated vesicles is not yet clear, although given that such vesicles contain TOM20, it is likely that they originated from the outer mitochondrial membrane. Alternative explanations for this mechanism include the delivery of cytoplasmic proteins from the ER rather than the cytosol, or potential pore leakage; these possibilities will require further investigation.

Aging and neurodegeneration can be partially characterized by an accumulation of protein aggregates and mitochondrial dysfunction[39]. In yeast, cytosolic proteins prone to aggregation are imported into mitochondria for degradation[16]. Mitochondria in aged or Dosmit-overexpressing muscle were found to exhibit an increased intramitochondrial ubiquitin signal, implying a defect in protein homeostasis (Supplementary Fig. 11). High expression of Dosmit resulted in lower locomotor activity and greater aggregation of ubiquitinated proteins (Supplementary Figs. 10 and 11), supporting the hypothesis that Dosmit may be involved in regulating mitochondrial health and transporting protein aggregates into mitochondria. In yeast, Heat shock protein 70 and protease are involved in mitochondrial unfolded protein degradation[16]. The exact counterparts in *Drosophila* and other higher organisms are unknown. Changes in mitochondrial content during aging, as well as Liquid chromatography–mass spectrometry (LC-MASS) analyses of mitochondrial protein content, should provide clues to the major players, however. Dosmit-mediated transport could provide an alternative route for mitochondria–cytosol communication and regulate protein uptake from the cytosol during aging.

There are two Dosmit homologs in mammals: Cisd1 and Cisd2[51,52]. These two proteins show different subcellular localization patterns and may have distinct functions, since Cisd2 has a longer N-terminal domain and is located mainly in the ER[63], while Cisd1 has a shorter N-terminal domain and is located mainly on mitochondria[51] (Supplementary Fig. 16b). The *D. melanogaster* homolog of Cisd, which is thought to be an earlier evolutionary form of this protein family, contains features of both Cisd1 and Cisd2. This implies that Dosmit may be functionally similar to both mammalian Cisd homologs. This protein family includes mammalian Cisd1–3[64]. *Drosophila* has only two CDGSH proteins: Dosmit (between Cisd1 and Cisd2) and CG3070 (Cisd3). The CDGSH domain appeared early in evolution[64]. Both the human and fly Cisd3 proteins contain two CDGSH domains, although whether Cisd3 plays similar roles to Dosmit requires further study. Further experiments in mammalian models may provide more evidence for the role of Dosmit in these animals. This conservation of transport mechanisms across species could make Dosmit a promising target for future therapies targeting aging and diseases related to mitochondrial dysfunction.

## Methods

**Fly stocks and maintenance**. Flies were maintained on standard fly food at 25 °C. The fly stocks used in our experiments were as follows: Mef2-GAL4 (Bloomington Stock Center (BSC), #50742), IFM-GAL4 (provided from Dr. Bruce A. Hay), *Dosmit-EP* (BSC, #30170), UAS-mito-GFP (BSC, #8442), UAS-YFP-Rab32[WT] (BSC, #9815), UAS-YFP-Rab32[CA] (BSC, #23280, Rab32[Q79L]), UAS-YFP-Rab32[DN] (BSC, #23281, Rab32[T33N]), UAS-GFP (BSC, #1521), Rab32[1] (BSC, #338), UAS-Tomboy20 RNAi (BSC, #67811), UAS-Tomboy40 RNAi (BSC, #29573), UAS-MGE RNAi (BSC, #57430), UAS-MCU RNAi (BSC, #42580), UAS-ND-39 RNAi (BSC, #52922), UAS-Tom7 RNAi (BSC, #79381), UAS-Tom20 RNAi (BSC, #64915), UAS-Tom40 RNAi (BSC, #26005), UAS-Tom70 RNAi (BSC, #64627), UAS-CG5662 RNAi (VDRC, #35030), UAS-CG9393 RNAi (VDRC, #44400), UAS-CG8443 RNAi (VDRC, #42136), UAS-Atg8a RNAi (VDRC, #109654), *Marf[E]* (BSC, #67155), *Opa1[s3475]* (BSC, #12188), *Drp1[T26]* (BSC, #3662), UAS-KDEL-GFP (ER retention sequence KDEL tagged with GFP, BSC, #30903), UAS-Golgi-GFP (Golgi localization sequence from the Homo sapiens B4GALT1 gene tagged with GFP, BSC, #30902), UAS-SPG7 RNAi (BSC, #57843), UAS-Rhomboid-7 RNAi (BSC, #51752), UAS-HtrA2 RNAi (BSC, #55165), UAS-dj-1beta RNAi (BSC, #38378), and UAS-Lon RNAi (BSC, #40162), UAS-Tom20-HA (FlyORF, #003545), UAS-Drp1 and UAS-Marf (provided from Dr. Yan-Shan Fang).

**Plasmids**. The plasmids for transgenic flies: UAS-Dosmit, UAS-Dosmit KD, Dosmit-2-1, Opa1-HA, Drp1-HA, Marf-HA, UAS-Dosmit-ΔN-HA, UAS-Dosmit-ΔCDGSH-HA, UAS-mCisd1-HA, UAS-mCisd2-HA, UAS-N-mCisd2-mCisd1-HA. The plasmids for cell transfection: pAc5.1-Rab32-HA, pAc5.1-Hsc70-5-HA, pAc5.1-Dosmit-HA, pAc5.1-mCisd1-HA, pAc5.1-mCisd2-HA, pAc5.1-N-mCisd2-mCisd1-HA, pcDNA3-mCisd1-HA, pcDNA3-mCisd2a-HA, pcDNA3-N-mCisd2-mCisd1-HA. The primer sequences used in this study are listed in Supplementary Table 1. The ectopic expression of protein was carried out in UAS transgenic flies. cDNAs were cloned into the pUAST-attB vector. The knocking down experiment of Dosmit is executed by micro-RNA-based knocking down mechanism as described in ref. [65]. Microinjection was performed by WellGenetics (Taipei, Taiwan).

**Antibodies**. Antibodies specific to the following substrates were used: phalloidin (1:1000, P1951; Sigma-Aldrich, St. Louis, MO, USA); ATP5A (1:10,000, #14748; Abcam, Cambridge, UK); Porin (1:10,000, #14734; Abcam, Cambridge, UK); GAPDH (1:10,000, #627408; GeneTex, Hsinchu, Taiwan); Tubulin (1:10,000, #628802; GeneTex, Hsinchu, Taiwan); GFP (1:10,000, #6556; Abcam, Cambridge, UK); HA (1:10,000, #3724; Cell Signaling Technology, Danvers, MA, USA), HA (1:10,000, #05-904; Merck Millipore) and Ubiquitin (1:500, #3933S, Cell Signaling Technology, Danvers, MA, USA). The Dosmit protein consisted of 133 amino acids, and Fis1 consisted of 154 amino acids. Dosmit and Fis1 antibodies were generated by immunizing rabbits with purified full-length Dosmit and Fis1.

**Immunofluorescent staining of *Drosophila* muscle**. Each fly thorax was dissected in 1× phosphate-buffered saline (PBS) and fixed with 4% paraformaldehyde (Sigma-Aldrich) for 30 min at room temperature. After washing with 0.1% Phosphate Buffered Saline with Triton X-100 (PBST) (Triton-X 100), the samples were transformed into a block with 2% bovine serum albumin (BSA) for 1 h at room

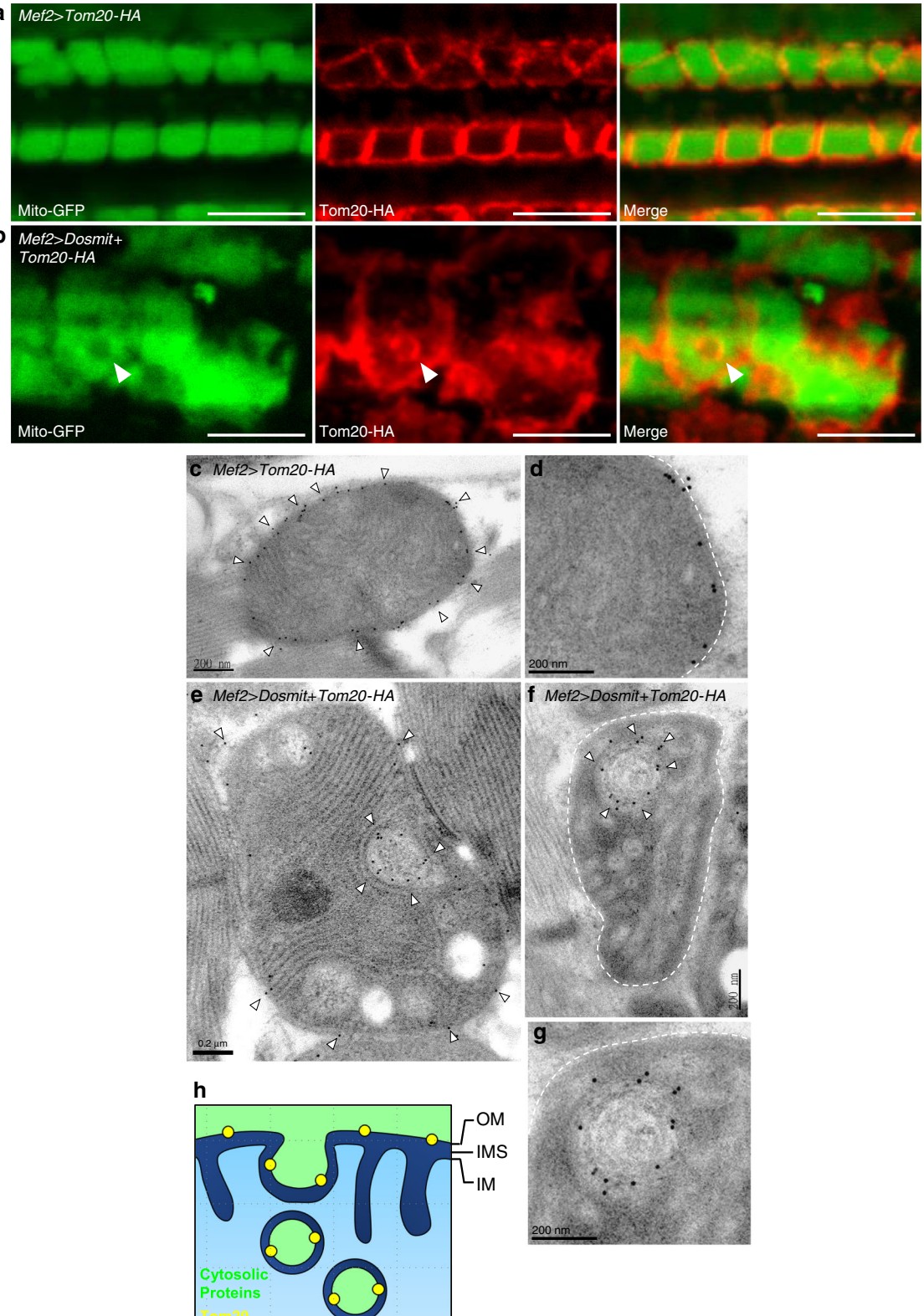

**Fig. 6 Ectopic expression of Dosmit induces the formation of intramitochondrial vesicles containing Tom20. a** Confocal microscopy images showing the HA-tagged outer mitochondrial membrane protein, Tom20-HA (red), localized on the outer region of mitochondria in flight-muscle tissue from *Mef2 > Tom20* flies in which mito-GFP (green) was also expressed. **b** In *Mef2 > Dosmit + Tom20-HA* flies that also expressed mito-GFP, Tom20-HA (red) localized on the membranes of intramitochondrial vesicles (arrowheads). Scale bar: 5 μm. **c** TEM images showing that gold-labeled Tom20-HA only localized on the outer mitochondrial membrane in the flight muscles of control flies (arrowheads). **d** Magnified image of **c**. **e**, **f** TEM images of gold-labeled Tom20-HA localizing primarily on the membranes of intramitochondrial vesicles (arrowheads) in the flight muscles of *Mef2 > Dosmit + Tom20-HA* flies in which Dosmit was ectopically expressed. Scale bar: 200 nm. **g** Magnified image of **f**. **h** Working model of cytosolic protein uptake and intramitochondrial vesicle formation (OM outer mitochondrial membrane, IMS intermembrane space, IM inner mitochondrial membrane).

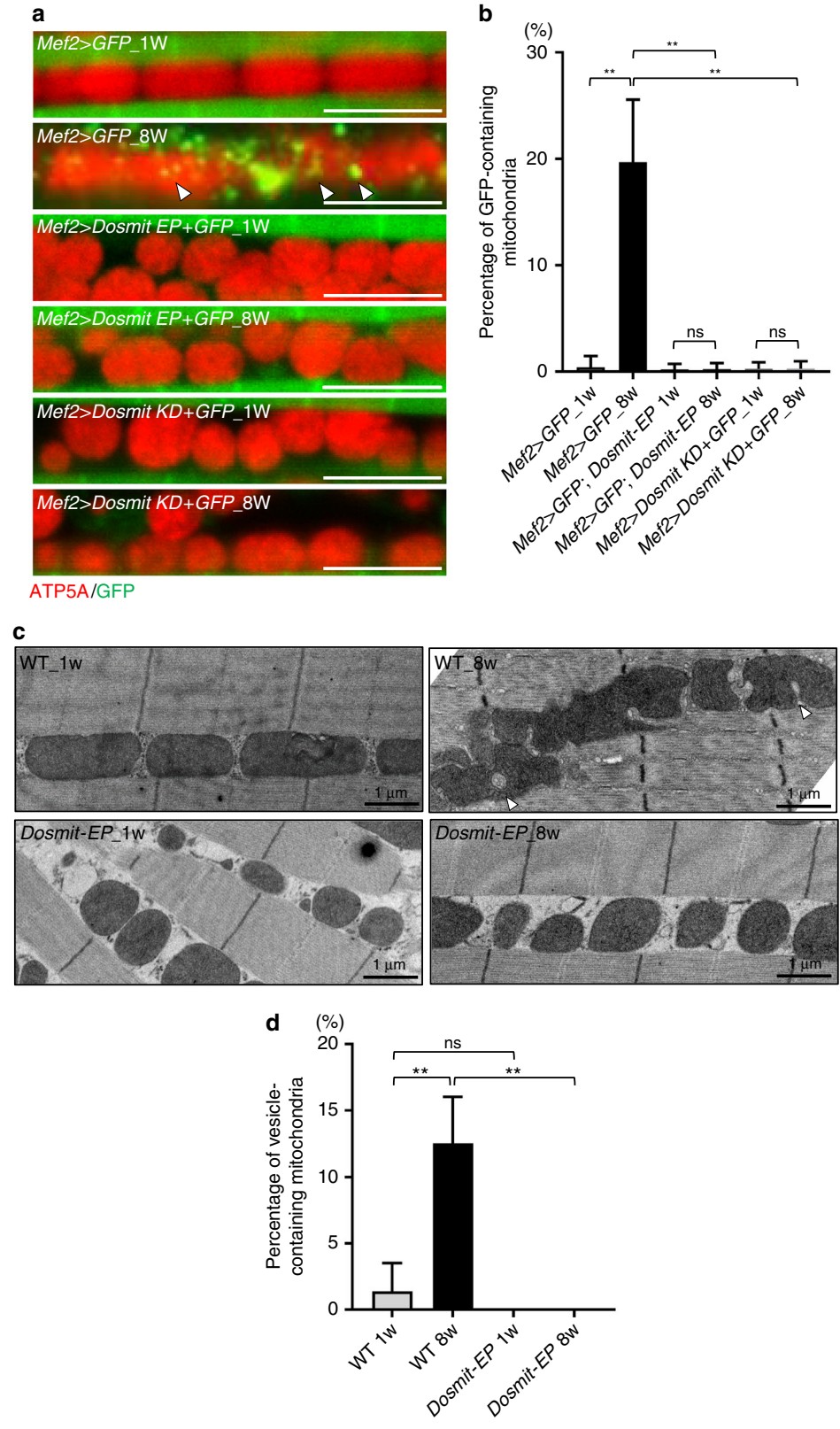

**Fig. 7 Aged mitochondria exhibit internalization of cytosolic content, but no vesicles appear in aging *Dosmit-EP* flies. a** Comparison of mitochondrial size in control (*Mef2 > GFP*) and *Mef2 > Dosmit-EP + GFP* and *Mef2 > Dosmit KD + GFP* flies (aged either 1 or 8 weeks) containing cytosolic GFP (arrowheads). Scale bars: 5 μm. **b** Percentage of wild-type mitochondria containing GFP in *Mef2 > GFP*, *Mef2 > Dosmit KD + GFP* and *Mef2 > Dosmit EP + GFP* flies aged 1 or 8 weeks (mean ± SD). *N* = 397, 340, 407, 400, 328, and 328 from left to right bars. Statistical test: Two-tailed Mann–Whitney *U*-test (\*\**p* < 0.01, ns: *p* > 0.9999). **c** TEM images of vesicle-containing mitochondria in wild-type and *Dosmit-EP* mutant flies (aged either 1 or 8 weeks) compared with a 1-week-old fly. **d** Percentage of vesicle-containing mitochondria in wild-type and *Dosmit-EP* mutant flies aged 1 or 8 weeks (mean ± SD). *N* = 143, 125, 162 and 169 from left to right bars. Statistical test: Two-tailed Mann–Whitney *U*-test (\*\**p* < 0.01; ns: *p* = 0.4545). Source data are provided as a Source Data file.

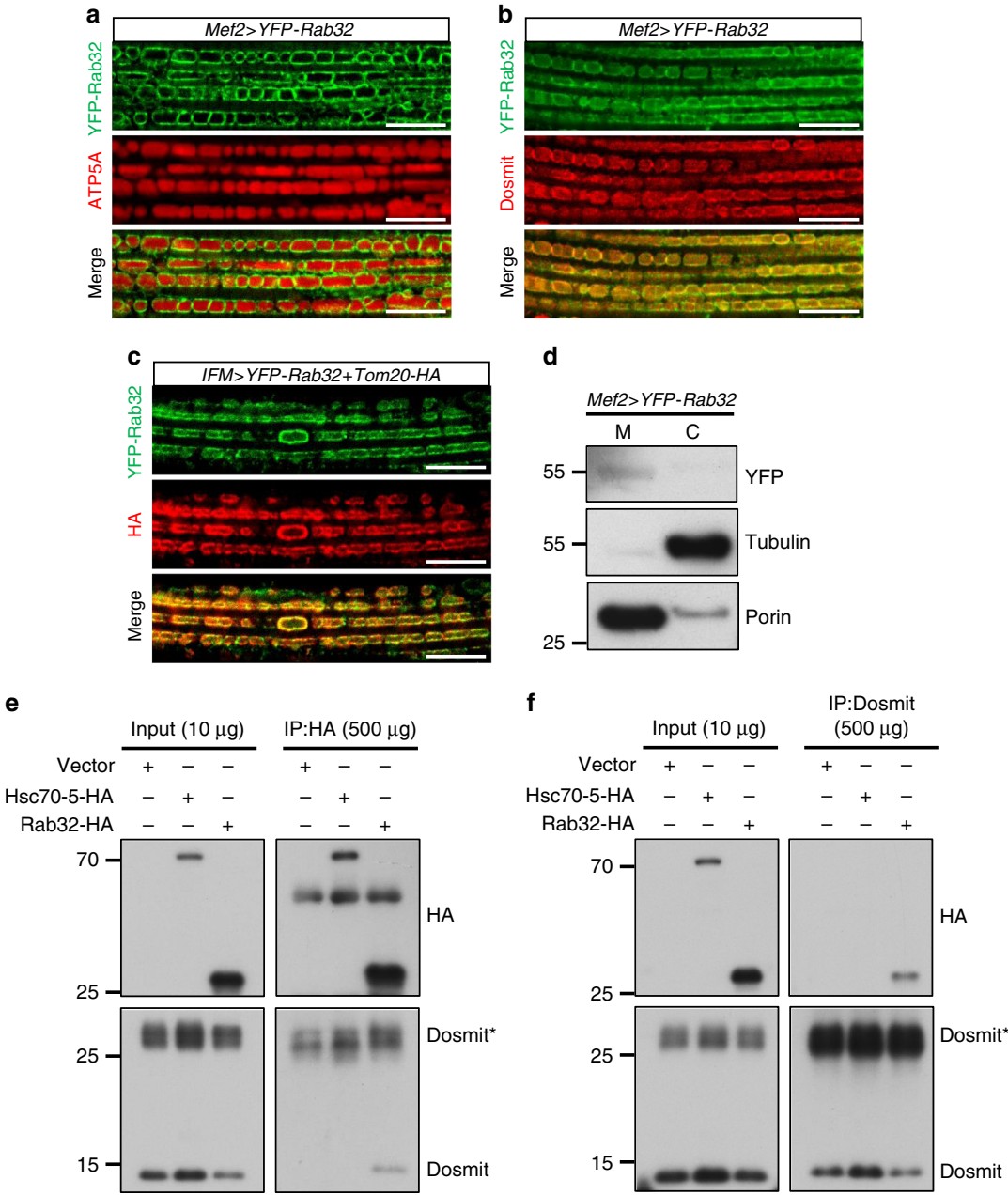

**Fig. 8 The mitochondria-associated protein, Rab32, colocolized with Tom20-HA and undergoes a protein–protein interaction with Dosmit. a** Confocal microscopy images of YFP-Rab32 (green) localized around mitochondria (red) and **b** colocalized with Dosmit (red) in fly muscle tissue. Scale bars: 10 μm. **c** Tom20-HA (red) ectopically expressed and colocalized with YFP-Rab32 (green). **d** Western blotting of flight-muscle tissue homogenates showed expression of the YFP-Rab32 protein in the mitochondrial fraction only (M: mitochondrial fraction; C: cytosolic fraction; Tubulin: cytosolic control; Porin: mitochondrial control). S2 cells transfected with Hsc70-5-HA or Rab32-HA plasmids. **e** When Rab32-HA or Hsc70-5-HA is immunoprecipitated with an anti-HA antibody, Dosmit can be detected in the Rab32-bound complex. **f** When endogenous Dosmit proteins are immunoprecipitated with the anti-Dosmit antibody, only Rab32-HA can be detected in the Dosmit–pulled-down complex. For the pull-down, 10 μg of input was loaded. For SDS-PAGE, 500 μg protein from the immunoprecipitation was loaded. Source data are provided as a Source data file.

temperature, and hybridized with primary antibodies overnight at 4 °C. After washing with PBST, the samples were incubated with secondary antibody–fluorophore conjugates overnight at 4 °C. The samples were washed again with PBST, the specimens were mounted in VectaShield Antifade Mounting Medium (Vector Laboratories, Burlingame, CA, USA), and examined using a Leica SP1 confocal microscope. Three-dimensional (3D) movie was performed by Zeiss LSM880 microscope (Objective LD LCI Plan-Apochromat 40×/NA1.2).

**Western blotting**. Each fly thorax was lysed with a mixture of RIPA buffer and sample loading buffer, and then boiled for 10 min. Samples were subjected to electrophoresis on a 10% acrylamide gel, and the resolved bands were transferred

on to a PVDF membrane. The membranes were then transformed into a block with 5% skim milk in Tris Buffered Saline with Tween-20 (TBST) for 1 h at room temperature, and then hybridized with primary antibodies overnight at 4 °C. After washing with TBST, the membranes were incubated with 1:10,000 dilution of HRP-conjugated secondary antibody in 5% skim milk for 1 h. Again after washing, the labeled proteins were visualized using a chemiluminescent HRP substrate (Milli-pore, Burlington, MA, USA).

**Lifespan assay**. One-hundred flies of each genotype were divided to five vials (20 flies/vial), and maintained in normal food at 25 °C. Dead flies were counted until all were dead. Before the lifespan assay, we backcrossed the *Dosmit-EP* mutant to

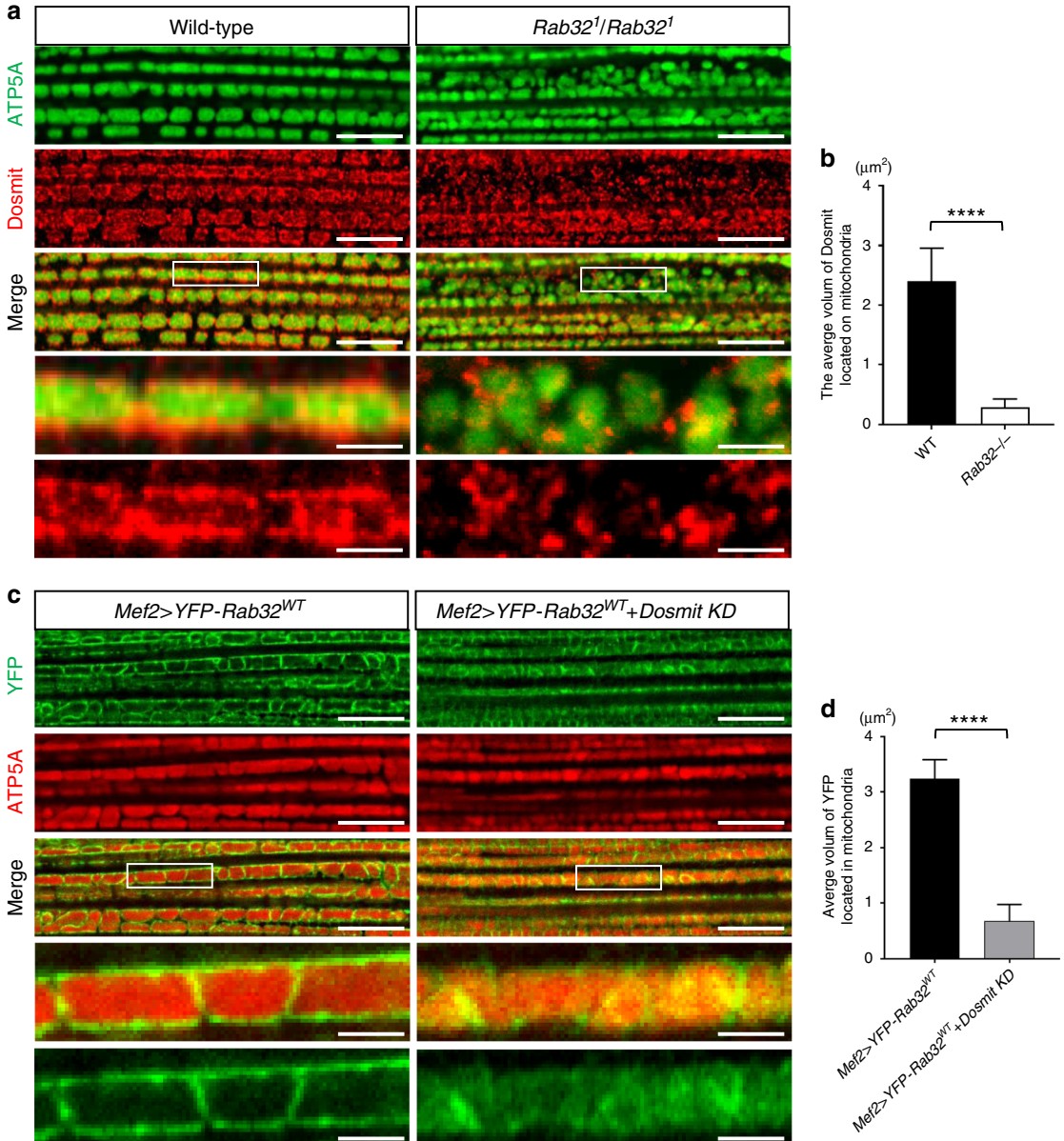

**Fig. 9 Rab32 and Dosmit localization on mitochondria is mutually dependent. a**, **b** Rab32 mutants show an improper localization of Dosmit compared to wild-type flies (mean ± SD). Dosmit: red; ATP5A: green. $N = 123$ and 215 from left to right bars. Statistical test: two-tailed student's $t$-test (****$p < 0.0001$). **c**, **d** Dosmit knockdown also significantly disrupted Rab32 localization (mean ± SD). YFP-Rab32: green; ATP5A: red. $N = 102$ and 191 from left to right bars. Statistical test: two-tailed student's $t$-test (****$p < 0.0001$). Scale bars: 10 μm. Source data are provided as a Source data file.

wild-type $w^{1118}$ for ten generations, and the CRISPR mutant *Dosmit-2-1* for five generations.

**Climbing assay**. The climbing assay was performed in a dark room with dim red light. Fifty flies that were not anesthetized with $CO_2$ in 24 h were collected for the assay. Ten test vials were first placed on the countercurrent apparatus and flies were loaded in the 1st vial in the transfer state. The flies were left 3 min to get acclimatized. Each genotype was dispelled to bottom of vial by gently tapping and transferred vials to testing state. The flies were left for 1 min to climb up. Transfer vials from testing state to transfer state and count the number of the flies in each test vial. Each genotype would be repeat for 3 times.

**Co-immunoprecipitation (Co-IP) assay**. Drosophila S2 cells were lysed with lysis buffer (50 mM Tris-HCl pH 8.0, 100 mM NaCl, 5 mM EDTA, 0.5% Nonidet P-40) containing 1x protease inhibitor (Roche, Cat. No. 04693116001). Cell lysates were centrifuged at 16,100 x $g$ for 10 min, and 500 μg of protein was incubated with 1 μg antibody at 4 °C for 12–16 h, and then incubated with 15 μl protein G Mag Sepharose beads (GE Life Sciences, 28967066) at 4 °C for 3 h. The immunoprecipitates were washed with lysis buffer three times, and sample buffer was added for SDS-PAGE analysis.

**Digitonin/proteinase K assay for S2 cells and tissue**. The cell medium was removed and the S2 cells were washed with KHM buffer (110 mM potassium acetate, 20 mM HEPES-NaOH, 3 mM $MgCl_2$, pH 7.2). Cells were incubated with 40 μM or 1600 μM digitonin in KHM buffer at room temperature for 20 min. After washing cells with KHM buffer, 50 ng/μl proteinase K was added to cells. After 15 min, 30 min, 60 min, 2 mM PMSF was added for 10 min on ice to stop protease K activity. The cells were then collected for immunofluorescence experiments or western blot assays. For tissue permeabilization, 1000 μM digitonin was applied for 30 min at room temperature, and the tissue was then treated with 100 ng/μl proteinase K for 2 h at 4 °C.

**Transmission electron microscopy**. Each fly thorax was dissected in 0.1 M cacodylate buffer and fixed in 1% glutaraldehyde and 4% paraformaldehyde in 0.1 M cacodylate buffer overnight at 4 °C. After washing with 0.1 M cacodylate buffer, the samples were treated with 1% $OsO_4$ for 1 h at 4 °C. After washing with ddH2O, the

samples were dehydrated in a gradient series using 30%, 50%, 70%, 95%, and 100% ethanol solutions at room temperature. The samples were then infiltrated using series epoxy resin at room temperature and mounted in pure resin. After polymerization at 60 °C for 2 days, 90 nm sections were prepared and observed with a Tecnai G2 Spirit TWIN transmission electron microscope (FEI, Hillsboro, OR, USA).

**Immunoelectron microscopy and immunolabelling.** Each fly thorax was dissected in 0.1 M cacodylate buffer and fixed in 0.1% glutaraldehyde and 4% paraformaldehyde in 0.1 M cacodylate buffer overnight at 4 °C. After washing with 0.1 M cacodylate buffer, the samples were stained with 1% uranyl acetate for 1 h at 4 °C. After washing with ddH$_2$O, the samples were dehydrated in a gradient series

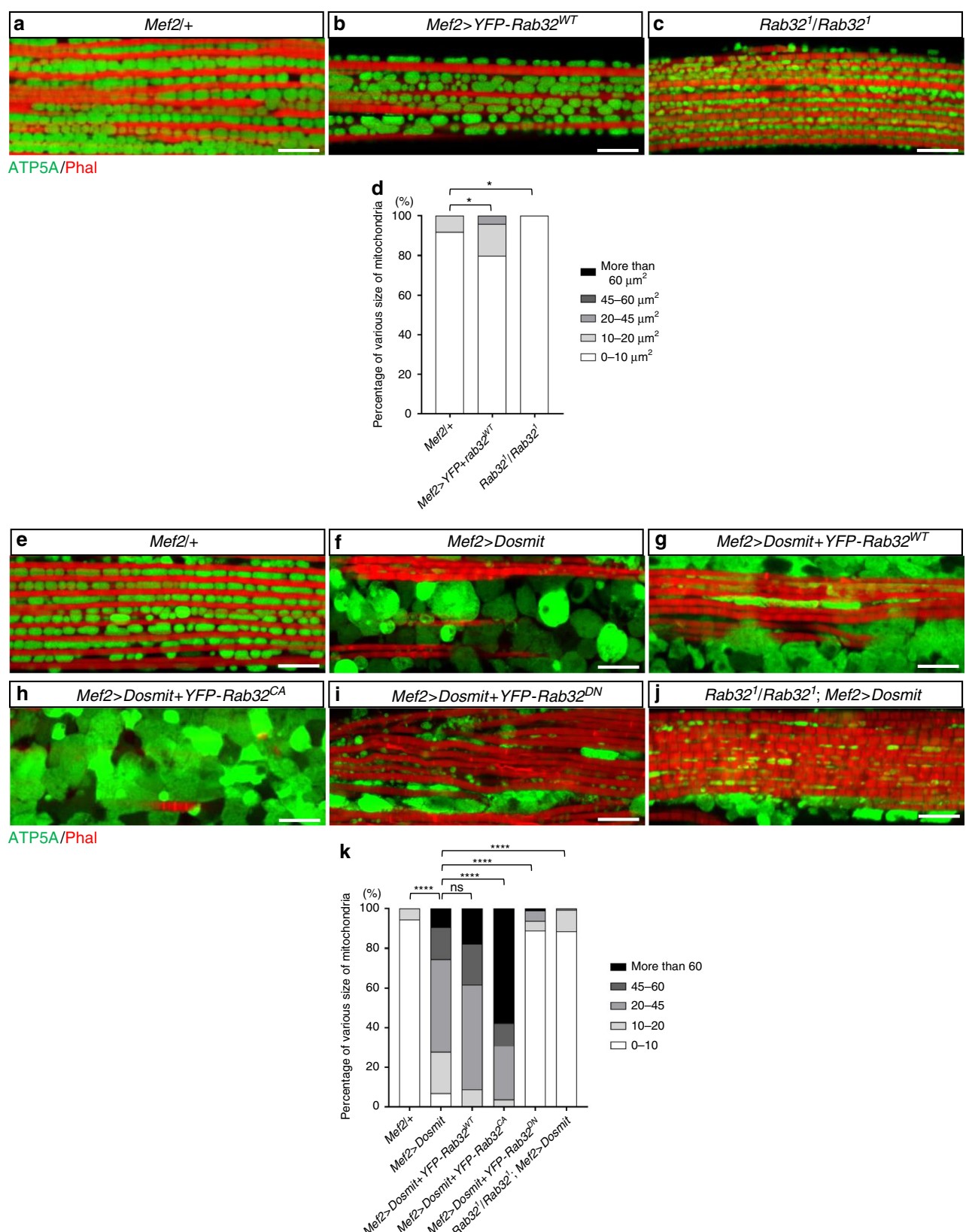

**Fig. 10** Continued

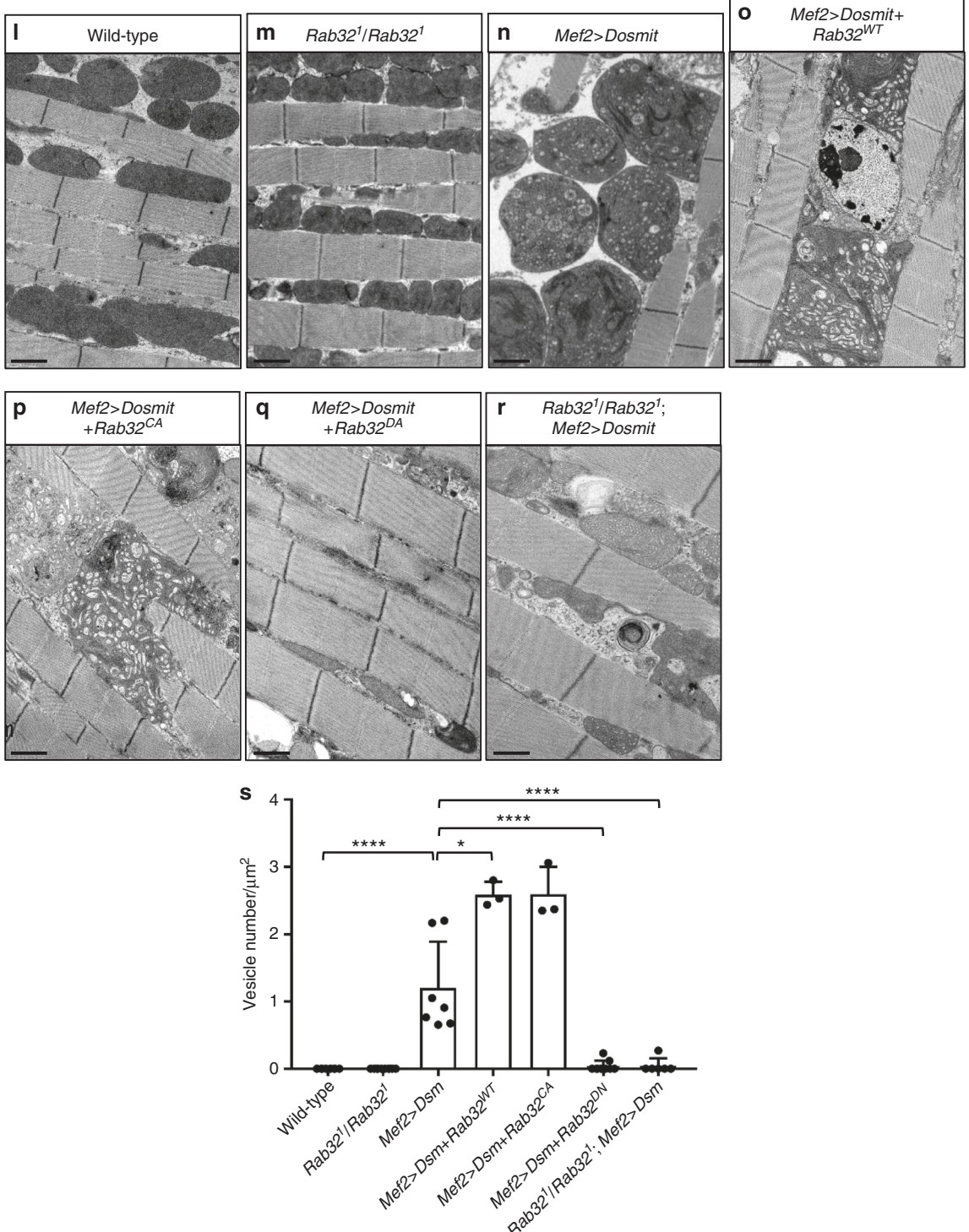

**Fig. 10 The mitochondria-associated protein Rab32 mediates the Dosmit-induced enlargement of mitochondria.** Mitochondria from: **a**, **e** *Mef2/+* control flies, **b** flies with ectopic expression of wild-type Rab32, **c** *Rab32¹/Rab32¹* flies, **f** *Mef2 > Dosmit* flies (in which Dosmit was ectopically expressed), **g** *Mef2 > Dosmit + YFP-Rab32^WT* flies (in which both Dosmit and wild-type Rab32 were ectopically expressed), **h** *Mef2 > Dosmit + YFP-Rab32^CA* flies in which both Dosmit and the constitutively active GTP-binding Rab32 mutant (Rab32^CA) were ectopically expressed, **i** *Mef2 > Dosmit + YFP-Rab32^DN* flies in which both Dosmit and the dominant-negative form of Rab32 (Rab32^DN) were ectopically expressed, and **j** *Rab32¹/Rab32¹; Mef2 > Dosmit* flies in which Dosmit was ectopically expressed in the Rab32-null mutant background. **d** Distribution of size classes of mitochondria in the various genetic backgrounds. $N = 62$, 124, and 115 from left to right bars. Statistical test: Chi-Square Test (*$p < 0.05$). Green channel shows ATP5A staining; red channel shows phalloidin staining of F-actin in muscle. Scale bars: 10 μm. **k** Distribution of size classes of mitochondria in the various genetic backgrounds. $N = 146$, 43, 34, 26, 126, and 131 from left to right bars. Statistical test: Chi-Square Test (****$p < 0.0001$; ns: $p = 0.0673$). **l–r** TEM images of mitochondria from wild-type, *Rab32¹/Rab32¹*, *Mef2 > Dosmit (Dsm)*, *Mef2 > Dosmit + Rab32^WT*, *Mef2 > Dosmit + Rab32^CA*, *Mef2 > Dosmit + Rab32^DN*, and *Rab32¹/Rab32¹; Mef2>Dosmit* muscle tissue. Scale bars: 2 μm. **s** Number of vesicles per unit area for 1-week-old flies from panels **l–r** (mean ± SD). $N = 8$, 10, 7, 3, 3, 9, and 6 from left to right bars. Statistical test: Two-tailed Mann–Whitney *U*-test (****$p < 0.0001$; *$p < 0.05$). Source data are provided as a Source data file.

using 30%, 50%, 70%, 95%, and 100% ethanol solutions at −20 °C. The samples were then infiltrated with series LR-gold and mounted in pure LR-gold. After polymerization by UV crosslinking, 90 nm sections were prepared and transformed into a block with 5% normal goat serum (NGS) for 1 h, before incubation with primary antibodies for 1 h at room temperature. After washing with 1% NGS, samples were incubated with secondary antibodies for 30 min at room temperature. Later they were again washed with 1% normal goat serum (NGS), the sections were then stained with 4% uranyl acetate and lead citrate before examination by TEM.

**Isolation of mitochondria from *Drosophila* flight-muscle tissue**. The mitochondria were isolated using the Mitochondria Isolation Kit for Tissue (Thermo-Fisher, Waltham, MA, USA), according to the manufacturer's protocol. The mitochondrial fraction definitely contains mitochondria, it may also potentially contain ER and other membranes.

**Statistics and reproducibility**. Mitochondrial volume was measured using ImageJ software. Statistical analysis and graphs was performed using GraphPad Prism 7.0. Significant differences were found between the data for different genotypes. For two group comparisons, either unpaired student's $t$-test (for parametric analysis) or Mann–Whitney $U$-test (for non-parametric analysis) were used. Comparison in mitochondrial distribution between the two genotypes is analyzed by chi-square test. (Significance levels: $*p < 0.05$, $**p < 0.01$, $***p < 0.001$, $****p < 0.0001$). ns, not significant. Every experiment in this study was repeated independently at least two times.

**Reporting summary**. Further information on research design is available in the Nature Research Reporting Summary linked to this article.

## Data availability

Please see https://doi.org/10.6084/m9.figshare.12116907 for source data. All the data that support the findings in this study are available from the corresponding author upon reasonable request. The source data underlying Figs. 1b, d, 2a–d, f, h–i, k, 3a–b, d, f, 4b, 5c–d, f, 7b, d, 8d, e, f, 9b, d, 10d, k, s, Supplementary Figs. 2b, 4b (bottom left panel), 4c, g–h, 5d, h, i. 5j, 6a, b, 7a–d, 8i, 9a, b, 10a, 11i–j, 12d, 13g, 14f, 16c, d, 17e, f, 18c, 19a, b, 20f, 21c, 22e are provided as a Source data file.

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

## Acknowledgements

We like to thank David Chan, Bruce Hay, Kah Leong Lim, Henry Sun for discussion for this manuscript. The fly lines were supplied from Bloomington stock center, FlyORF stock center, Taiwan flycore. Also thanks Ting-Fan Tsai for sharing reagents. T.E.M. was supported from IMB, Academia Sinica, and Technology Commons, College of Life Science, National Taiwan University. The grant support from MOST 105-2633-B-400 -001, NHRI grant, MR-107-PP-09.

## Author contributions

P.L.C., K.T.H., J.C.L., C.H.C. designed research; P.L.C., K.T.H., C.Y.C., H.Y.C., J.C.L., T.Z.L. performed research; and P.L.C., K.T.H., J.C.L., M.P.S., W.Y.Y., H.C.C., and C.H.C. analyzed data; and P.L.C., M.P.S, and C.H.C. wrote the paper.

## Competing interests

The authors declare no competing interests.

## Additional information

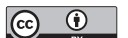

