## [Peer Review File · Nature Communications]

Reviewers' comments:

Reviewer #1 (Remarks to the Author):

Mitochondria constantly fuse and divide in a dynamic tubular network. The mitochondrial dynamics is well controlled by specific factors described for the outer and inner membrane. Chen and colleagues report that mitochondrial size increases with age in *Drosophila melanogaster*. In parallel, increasing number of vesicles has been observed in mitochondria with aging that might contain cytosolic content. They could identify a new component termed Dosmit that is linked to the increased size of mitochondria during aging. Depletion of Dosmit blocks increasing size of mitochondria and intramitochondrial vesicle formation. The data pointing to a link of Dosmit to mitochondrial dynamics are convincing. However, mechanisms how Dosmit affect mitochondrial dynamics remain unclear. Furthermore, origin and function of intramitochondrial vesicles remain unclear. At the present stage, the manuscript presents interesting observations, but fails to provide mechanistic insights.

Major points:

1. The authors reported that the increasing mitochondrial size during aging depends on Dosmit (Figure 2). However, the molecular mechanisms remain obscure. Are levels of protein known to control mitochondrial dynamics like Fis1, Mfn1, Opa1, Marf1 and Drp1 in Dosmit mutant cells affected?
2. Chen and colleagues propose that Fis1, Opa1, Marf1 are not important to control mitochondrial size. However, the steady state levels of Marf1 and Opa1 decreases during aging, while more Drp1 and Fis1 were observed, indicating that these proteins play an important role in this process. The authors have to clarify this point by depleting these components and analyze mitochondrial size during aging. Does depletion of mitofusins that are important for mitochondrial fusion affect mitochondrial size?
3. The authors report that vesicles are formed inside mitochondria. The function of these vesicles remains unclear. The authors propose that these vesicles mediate import of mitochondrial proteins that lack a canonical mitochondrial targeting signal. However, no experimental data are provided that support this conclusion. Could further proteins such as mitochondrial preproteins be detected in these vesicles? Does block of vesicle formation affect mitochondrial protein content (e.g. Rab32 mutant)? In this context, many mitochondrial proteins lack a cleavable mitochondrial targeting signal and are imported via specific protein translocases, which has to be mentioned in the text.
4. Author report a role of Rab32 in the formation of intramitochondrial vesicles. How do Dosmit and Rab32 cooperate to mediate formation of intramitochondrial vesicles?

Minor points:

1. The authors report that Dosmit and Rab32 can be detected in a mitochondria-enriched fraction (Figs. 2h, 5c and 8c). Since mitochondrial preparations are often contaminated with ER-localized proteins, ER marker proteins and an ER fraction should be shown as control.
2. Do Dosmit and Rab32 localize to the mitochondrial outer membrane? This issue should be addressed by performing proteinase K treatment of intact and swollen mitochondria. Furthermore, do these proteins integrate into mitochondrial membranes?
3. The authors should describe the screen in which Dosmit was originally identified. How many different genes were tested and were identified?
4. The authors used a Dosmit mutation (Dosmit-EP) to analyze the role of the protein for mitochondrial morphology. The authors should describe that mutant in more detail.

Reviewer #2 (Remarks to the Author):

Chen and colleagues describe the identification of Dosmit (also known as Cisd2) in flies regulates mitochondrial morphology with age and present data that suggests it may do so by promoting the uptake of vesicles into mitochondria. This is a very interesting premise as only a very few mechanisms of mitochondrial uptake are known, and this would be a new one. As such, it is important to ensure such claims are correct. The data presented include a number of interesting observations, most notably, the presence of multiple intramitochondrial vesicle-like structures and the apparent uptake of cytosolic proteins, when Dosmit is overexpressed. While data are presented that supports a mechanistic role for Rab32 in this, it seems unclear exactly what purpose this process may serve. One of the more striking observations is that loss of Dosmit, via an EP element, and presumably loss of vesicle uptake, appears to lead to a substantial increase in lifespan and vice versa, although these data are unfairly relegated to supplementary information. Is this the same for 2-1 mutants? I'd like to know more about the organismal phenotypes of Dosmit mutant and OE (or RNAi). Are they more active into old age? Do they suppress markers of ageing?

Overall, this is an intriguing study but I'd like to know more about these vesicles, e.g. see more examples of the double-membrane bound vesicles (see more notes below). Determining the 'content' of these vesicles is extremely important to understanding this process. The cytosolic GFP experiment is a good start but some results don't align well. For instance, the fluorescence imaging, there appears to be strong re-localisation of GFP coincident with the mitochondrial signal. One oddity is where have all the myofibres gone? Is this representative? It is already slightly odd that the 'cytosolic' GFP seems to colocalise with the myofibres in the control condition, but I appreciate that there is hardly any 'cytosol' in this tissue. More importantly, the fractionation assay in Fig. 5c shows only a small proportion of the GFP re-localises into the mitochondrial fraction. This also doesn't seem to reflect the immuno-EM quantification, since this argues that ~70% of the molecules are in the mitochondrial vesicles. It would also be useful to have a sense of how well the immuno-EM detects the cytosolic GFP since hardly any is visible in these images so it is hard to know how representative the quantification in e is. Here, it would be useful to know the relative distribution of GFP-gold in control samples. Currently, it isn't clear how much of the cytosolic GFP re-localises. It should also be remembered that this is under transgenic overexpression. Also, that co-localisation of sub-cellular structures by light microscopy is difficult. What is the optical thickness of the confocal sections taken here?

Another very important set of observations is the suppression of age-related vesicles by Dosmit knockdown or mutation (Fig. 7), but why was the RNAi used for one assay and the EP mutant for another. It would be good to verify and extend these observations by repeating the assays with multiple mutant conditions.

The authors argue that the intramitochondrial vesicles are double-membrane bound, possibly having originated as invagination of the outer mitochondrial membrane. The cartoon in Fig. 6h suggests that the authors envisage that the vesicles also include inner membrane; (i) this idea can and should be tested using an IMM marker (ATP5A should do), and (ii) this would imply that they are triple-membrane bound. Is this observed?

It would be appropriate to know more about the potential mitochondrial localisation of Dosmit. The authors show mitochondrial enrichment in a crude fractionation but a greater depth is warranted especially if this mechanistically implicates cooperation with Rab32 outside mitochondria. I.e. OMM localisation could be shown by protease/digitonin assays, carbonate extraction would determine the

relative intergration etc. For Fig. 8 d-g, I would appreciate seeing the split channels of the high mag images to better appreciate the distribution of Rab32 and Dosmit in the different conditions.

Overall, the writing is reasonable but could do with a thorough review to improve the flow and accuracy of the text and data, and the current Abstract and Discussion are rather disjointed and rambling.

Minor:

In general, the labels on the immunofluorescence micrographs are hard to read and should be changed. Also, the indexing to the respective data figures could be clearer. And the order of Supplementary data seems to be quite random, e.g. Supp 4 follows Supp 5 in the text.

Fig. 1c. these images are too small and should be enlarged (also the corresponding chart). Besides the scale bars are similar in size but one reads 500 μm and the other 500 nm. One must be wrong.

It isn't clear if a HA knock-in was made for Fis1. The text suggests so but there no schematic in Supp. Fig. 1, and the Fig. 2d suggests that a Fis1 antibody was used (and not anti-HA) but none is listed in Methods. There is also a lack of consistency in the Supp figures whether the gene names are capitalised or not. (Follow FlyBase nomenclature).

Lines 186-7: "a Dosmit-EP mutant... suggesting that Dosmit may use Cys-Cys-linked dimer formation". This needs more explanation.

Fig. 3a-b. Is the quantification for total Dosmit or just one species?

Supp Fig. 4. The results are clear but for completeness a statistical analysis should be applied. Also, reporting lifespan parameters usually use median rather than mean.

Fig. 4a. The scale bars don't seem to correspond well between high and low mag images, especially the larval samples. Please check.

The data showing the presence of these vesicles in Atg8 knockdown should be shown and explained in details, e.g. which gene (Atg8a or b) was targeted? This also leads to a potential issue with the experiment since both genes would need to be targeted in order to block autophagy.

Line 267: "electronically dense" would be better expressed as 'electron-dense'.

Lines 267-8: I do not understand the sentence "we found a reduction in electronically dense material within these vesicles compared to control muscle tissue". What are the authors comparing exactly? There are no/few vesicles in the control tissue. Are they trying to compare with cytosol?

Line 315: "Small-membrane vesicles". What is a 'small-membrane'? Do the authors simply mean small, membrane-bound vesicles? If so, they could just say 'small vesicles' (which, by definition, are membrane-bound).

Line 391: "Rab32 was found in abundance in the mitochondria". This claim seems like an overstatement, and should be revised, given the rather faint signal shown.

Lies 388-9. The authors seem to suggest that they did a screen of Rabs but don't show this, which seems unnecessary and/or incomplete.

Lines 408-9: What do the authors mean by "homozygous knockdown mutants". Do they mean simply 'homozygous mutants'? They should also standardise their genetic nomenclature. Here description of genes/transgene/mutants should be Rab32.

Lines 428-39: This section is rather incongruous. First, it disrupts the flow placed here, but moreover, it doesn't specifically address the idea that the authors claim. It is interesting to know that Drp1 overexpression can reverse the Dosmit overexpression mitochondrial enlargement but this in no way implicates mitophagy in the process or phenotype.

Similarly, the next section of Dosmit structure-function disrupts the flow and would be better placed in the initial description of Dosmit overexpression phenotypes.

Lines 451-454: "we found that the N-terminus of Dosmit was also required for the enlarged-mitochondria phenotype (Supplementary Fig. 7d-f), suggesting that iron-sulfur binding may play an important role in regulating mitochondrial dynamics." The first part of this sentence does not justify the second part.

Line 455-6: "our data suggests that Dosmit-induced transport of vesicles into mitochondria which is dependent upon both Rab32 and Drp-1". They haven't shown that vesicle transport is dependent on Drp1, only the enlargement phenotype.

Lines 465-467: "Using an RNAi screen, we showed that the major mitochondrial dynamic components Opa1, Drp1, Marf, and Fis1 are not involved in vesicle formation". This was not shown, only mentioned in the text. Edit.

Lines 479-80: "are involved in subcellular localization". What does this mean?

Line 498: "(MDVs) have been reported to shuttle proteins between the cytosol and mitochondria". Not exactly as stated. They have been shown to transport mitochondrial components to other parts of the cell. This statement implies a reverse transport direction which is not currently claimed.

Reviewer #3 (Remarks to the Author):

In this paper, Chen et al. describes the phenotypes of Dosmit (aka *cisd2*) expression and knockout in *Drosophila*. Dosmit is an iron-sulfur binding protein and was identified to reduce the age-related size increase of mitochondria when knocked down. Conversely, upon expression of Dosmit, mitochondria contained double-membraned vesicles that appeared to transport cytosolic cargo and were positive for the outer mitochondrial membrane protein TOM20. The mitochondrial localization of Dosmit is dependent on Rab32 and Rab32 knock-out also leads to smaller mitochondria similar to Dosmit knockout. Protein levels of Dosmit and intramitochondrial vesicle formation increase with age of the flies. The authors conclude that Dosmit induced transport of vesicles might represent a therapeutically interesting novel route for mitochondrial protein trafficking and that this pathway might explain age-related mitochondrial changes.

While the appearance of double-membraned vesicles importing cytosolic components into mitochondria is very interesting and certainly novel, the manuscript seems largely descriptive. It is unclear how Dosmit and Rab32 lead to the formation of intramitochondrial vesicles, what their functions are, if any, and what functional consequences the vesicles have. In addition, it is unclear if the presence of double membrane intramitochondrial vesicles is fly specific or a universal mechanism that also applies to higher organisms. The authors point out that Dosmit knockout has a beneficial effect on mitochondrial morphology and longevity in *Drosophila*, but in mouse models *cisd2* knockout shows quite the opposite effect and recessive mutations in this gene are associated with a severe neurodegenerative disease (Wolfram syndrome) in humans. This leads to the question if the function of Dosmit is conserved in higher organisms. If not this narrows the scope of the presented work and it might be suited better for a more specialized audience. It is ultimately up to the editor to decide whether publication in *Nature communications* is justified.

Major points:

- As stated above, a major issue is that this phenotype seems to be limited to *Drosophila*
- Can the *dosmit* knockout phenotype(s) be rescued by expression of human *cisd1* or 2?
- Adequate description of mitochondrial morphology quantification is lacking. The authors should provide a detailed description for each of the assessed parameter and the samples size(s).
- Figure 3: Does the increase of *Dosmit* reflect the "increase in mitochondrial size within muscles as the flies aged" (line 129) and do other mitochondrial proteins increase with age, too? A mitochondrial loading control (*ATP5A* or *Porin*) should be added to exclude this possibility.
- Supplementary Fig.4: Why is the analysis of longevity only performed in male flies? Is there a statistical assessment?
- Figure 6: the depicted model for vesicle formation shows that the inner mitochondrial membrane forms the outer membrane of the vesicles. The authors should support this model experimentally by co-localizing inner mitochondrial markers to the vesicles.
- The authors mention that a small scale RNAscreen led to the identification of *Dosmit* and disqualified other candidates but the data is not shown.
- Line 282: The authors state that the uptake of cytosolic components constitutes a novel route of communication between the cytosol and the mitochondria, but it does not seem that there is any specificity for the uptake nor that there is a release of the vesicular components. As such, it is not clear how this communication works and what it controls.
- Line 388: " Rab protein screening (using *Rab32* as a target)". Please explain.
- The dependence of *Dosmit* and *Rab32* on each other regarding their mitochondrial localization is intriguing. It would be good to strengthen the data using biochemical methods and interesting to know if *Dosmit* and *Rab32* (directly) interact with each other.
- Figure 9: The authors state that the effect of *Dosmit* expression was partially suppressed in flies coexpressing the dominant negative *Rab32* or in *rab32* null mutant flies but this is not really the case. To suppress the phenotype would mean that the morphology should go back to normal however there are gross abnormalities visible in Fig. 9i-k. How do the authors explain this?

Minor points:

- In the abstract, line 23/24, the authors state that the *Dosmit* protein forms vesicles. Please rephrase.
- *Dosmit* EP mutant is mentioned before it is introduced as transposon mutant (line 186/187 and 189/190)
- Line 391: "in abundance" should be deleted or a strong

**Response to Reviewers' comments**

**Reviewer #1 (Remarks to the Author):**

***Mitochondria constantly fuse and divide in a dynamic tubular***
***network. The mitochondrial dynamics is well controlled by specific***
***factors described for the outer and inner membrane. Chen and***
***colleagues report that mitochondrial size increases with age in***
***Drosophila melanogaster. In parallel, increasing number of***
***vesicles has been observed in mitochondria with aging that might***
***contain cytosolic content. They could identify a new component***
***termed Dosmit that is linked to the increased size of mitochondria***
***during aging. Depletion of Dosmit blocks increasing size of***
***mitochondria and intramitochondrial vesicle formation. The data***
***pointing to a link of Dosmit to mitochondrial dynamics are***
***convincing. However, mechanisms how Dosmit affect***
***mitochondrial dynamics remain unclear. Furthermore, origin and***
***function of intramitochondrial vesicles remain unclear. At the***
***present stage, the manuscript presents interesting observations,***
***but fails to provide mechanistic insights.***

Thank you for your comments, we hope to address them in our detailed
response.

***Major points:***

***Q1. The authors reported that the increasing mitochondrial size***
***during aging depends on Dosmit (Figure 2). However, the***
***molecular mechanisms remain obscure. Are levels of protein***
***known to control mitochondrial dynamics like Fis1, Mfn1, Opa1,***
***Marf1 and Drp1 in Dosmit mutant cells affected?***

33 A: Our results show that wild-type mitochondria enlarge throughout the
34 aging process, and that the Dosmit expression level increases
significantly during this time (Fig. 3a–d). On the other hand, the
mitochondria of the Dosmit-mutants remained the same size even in old
age (Fig. 3e, f). This suggests that Dosmit is associated with
mitochondrial enlargement. The expression levels of Marf and Opa1
changed only slightly over the same period, but those of Fis1 and Drp1
changed more substantially. To address the reviewer's question, we
examined the levels of these four proteins in the Dosmit RNAi
background, and found that the changes in level of all of them were
highly similar in wild-type and Dosmit-RNAi flies (line# 237- 241) (Fig
Reviewer #1 Q1 /Sup Fig 7), suggesting that Dosmit-regulated
mitochondrial enlargement during aging may not depend on these
mitochondrial dynamic proteins. However, Dosmit-regulated

mitochondrial enlargement was reduced by high levels of Drp1
(Supplementary Fig. 5d), suggesting that there is nevertheless a link
between these two proteins. Detailed mechanistic mapping of this
protein interaction would be an important task of future research.

Fig Reviewer#1 Q1 /Sup Fig 7

The protein level of Marf, Opa1, Drp1, and Fis1 during aging is independent of Dosmit.

a–d) The patterns of Marf (a), Opa1 (b), Drp1 (c), and Fis1 (d) protein levels in Dosmit-knockdown flies during aging were not significantly different from those in Mef2/+ flies.

Q2. Chen and colleagues propose that Fis1, Opa1, Marf1 are not important to control mitochondrial size. However, the steady state levels of Marf1 and Opa1 decreases during aging, while more Drp1 and Fis1 were observed, indicating that these proteins play an important role in this process. The authors have to clarify this point by depleting these components and analyze mitochondrial

**size during aging. Does depletion of mitofusins that are important**
**for mitochondrial fusion affect mitochondrial size?**

72 A: We have not claimed that Fis1, Opa1, and Marf1 are not important in
controlling mitochondrial size; rather, we propose that mitochondrial
size change during aging is not totally dependent on these known
dynamic proteins. In order to resolve this question, as the reviewer has
suggested, we checked the mitochondrial morphology of Marf, Opa1,
and Drp1 mutants (line# 241-245) (Fig Reviewer #1 Q2 /Sup Fig 8).
Since these mutations are homogenous lethal, we used heterozygous
mutants of *opa1^{s3475}* and *marf^E* flies. They exhibited mitochondrial
fragmentation as normal (Supplementary Fig. 8c, e, i, as shown below),
and *drp1^{T26}* exhibited mitochondrial fusion in young flies
(Supplementary Fig. 8g, i, shown below). However, our results also
showed that mitochondria still became enlarged in *marf^E*, *opa1^{s3475}*, and
*drp1^{T26}* heterozygous mutants during aging (Supplementary Fig. 8d, f, h,
shown below), indicating that mitochondrial enlargement occurs even
when these mitochondrial dynamic proteins are blocked. Dosmit
therefore seems to have a significant role in aging-dependent
mitochondrial enlargement.

Fig Reviewer#1 Q2 /Sup Fig 8

The size of mitochondria still enlarged in aged *drp1^{T26}*, *opa1^{s3475}* and *marf^E* fly lines.

a, b) Wild-type flies exhibited mitochondrial enlargement with aging. **c–f)** One-week-old *opa1^{s3475}* and *marf^E* flies exhibited more fragmented mitochondria than wild-type flies did, but their mitochondrial size had increased by the eight-week-old stage. **g, h)** One-week-old *drp1^{T26}* exhibited more fused mitochondria than wild-type flies did, and their mitochondrial size had increased by the age of eight weeks old. Scale bars: 10 μ m. **i)** Categorization of mitochondrial size for one- and eight-week-old wild-type, *opa1^{s3475}*, *marf^E*, and *drp1^{T26}* flies. n=80-100 from 5–7 flies

**Q3. The authors report that vesicles are formed inside**
**mitochondria. The function of these vesicles remains unclear. The**
**authors propose that these vesicles mediate import of**
**mitochondrial proteins that lack a canonical mitochondrial**
**targeting signal. However, no experimental data are provided that**
**support this conclusion. Could further proteins such as**
**mitochondrial preproteins be detected in these vesicles? Does**
**block of vesicle formation affect mitochondrial protein content**
**(e.g. Rab32 mutant)? In this context, many mitochondrial proteins**
**lack a cleavable mitochondrial targeting signal and are imported**
**via specific protein translocases, which has to be mentioned in the**
**text.**

109 A: Knowing the content of these vesicles would certainly provide clues
to understanding their function. However, it has taken researchers
decades to understand the content of transport vesicles like exosomes.
We have tested several proteins, including ribosome subunits Rsp-2,
Atg8, and Ref(2)p, and they exhibit varying capacities for mitochondrial
entry via these intramitochondrial vesicles. We agree that there is
probably a rule for cargo selection, which would indeed be a good topic
to investigate in future research (Fig Reviewer #1 Q3-1).

In this revised version, we do identify one important type of molecule
contained by these vesicles: ubiquitinated proteins. We have included
these results as (line# 423-436) (Fig Reviewer #1 Q3-2/ Sup Fig 12).

There is a report illustrating that mitochondria are involved in
protein homeostasis, including the processing of unfolded proteins¹. The
striking experimental data we have added to the manuscript suggest
that these intramitochondrial vesicles may be involved in the uptake of
ubiquitinated proteins. Specifically, we showed that ubiquitinated
proteins accumulated in the vesicles of Dosmit-induced enlarged
mitochondria and aged mitochondria, suggesting these
intramitochondrial vesicles function in transporting ubiquitinated proteins
into mitochondria(line#423-436), (Fig Reviewer #1 Q3-2/ Sup Fig 12a–
130 d, i, j, shown below). Further LC/MASS experiments may explain
differences in the cargo of these vesicles in the future.

In general, the TOM/TIM pathway is responsible for the transport of
mitochondrial proteins containing a cleavable target signal. Although
this new route we have identified may be an alternative pathway for the
import of mitochondrial proteins, it is as yet unclear whether this is the
case, since we do not yet have enough related results. It is therefore
also not clear whether mitochondrial preproteins exist in
intramitochondrial vesicles.

We also do not yet know whether mitochondrial content is affected
if vesicle formation is blocked. We attempted to isolate mitochondrial
proteins to conduct a proteomics-based study, in order to compare the

protein content of *rab32¹/rab32¹* mitochondria with that of controls

Fig Reviewer#1 Q3-1

Cytosolic ribosome subunit Rps2 proteins could be delivered into mitochondria, but not Ref(2)p and Atg8a.

a) Mef2> Rps2-HA b) Mef2> Dosmit+Rps2-HA, a, b) stained with HA and ATP5A antibody. c) Mef2/+ and d) Mef2> Dosmit muscle stained with ATP5A and Ref(2)p antibody, Ref(2)p signals enhanced when Dosmit highly expressed. e) Mef2> mcherry-Atg8a and f) Mef2> Dosmit, mcherry-Atg8a stained with ATP5A. Only ribosome subunit Rps2 proteins could be delivered into mitochondria. Scale bar: 10 μ m.

Fig Reviewer#1 Q3-2 /Sup Fig 12

Ubiquitinated proteins accumulate within the mitochondria in ectopically expressed Dosmit flies and aged wild-type flies.

a, b) Mef2>mitoGFP (**a**) and Mef2>Dosmit+mitoGFP (**b**) fly muscle stained with ATP5A (green) and ubiquitin antibody (red). Ubiquitinated proteins accumulated in the mitochondria of Mef2>Dosmit+mitoGFP flies. **c, d)** Ubiquitinated proteins also accumulated in large puncta and were distributed throughout aged wild-type muscle, particularly in the intramitochondrial vesicles (arrowheads). **e–h)** Ubiquitinated proteins did not form or accumulate in either young or aged Dosmit-EP or Dosmit-2-1 flies. Scale bars: 10 μ m. **i, j)** Quantification of mitochondria in which ubiquitinated protein accumulated. n=80-100 from 4–6 flies.

**Q4. Author report a role of Rab32 in the formation of**
**intramitochondrial vesicles. How do Dosmit and Rab32 cooperate**
**to mediate formation of intramitochondrial vesicles?**

In the original version of this paper, we demonstrated a genetic
interaction between Dosmit and Rab32: while gain-of-function mutation
of the constitutively active form of Rab32 GTPase enhanced
mitochondrial enlargement, loss-of-function mutation (whether a
dominant-negative GTPase mutant or a GTPase-null mutant) greatly
reduced the phenotype. We also showed that Rab32 is associated with
mitochondria and mutually dependent on Dosmit for maintenance.
To address the question of how Dosmit and Rab32 cooperate to
mediate the formation of intramitochondrial vesicles in more detail, we
performed TEM to examine the correlations in previous genetic assay
data in detail (line #494-497) (Fig Reviewer #1 Q4-1 /Sup Fig 10 l-s)
indicates that there is a strong correlation between the activity of Rab32
enzyme and Dosmit in connection with intramitochondrial vesicles.
We also conducted an immune pull-down assay to investigate whether
there is a physical interaction between these two proteins (Line # 456-
460)(Fig Reviewer#1 Q4-2 / Fig 8 c, e, f) confirms that there is such an
interaction. These lines of evidence provide the insight that these
proteins cooperate to bring about vesicle formation.

It has been reported that Sorting nexin 6 (SNX6) is one effector
target of the Rab32 GTPase in mammals. Which protein is the effector
of *Drosophila* Rab32 and how they function together with Dosmit and
the Rab32-SNX protein complex to induce membrane curving will be
investigated in the near future.

Fig Reviewer#1 Q4-1 /Sup Fig 10 l-s

**The mitochondria-associated protein Rab32 mediates the Dosmit-
induced enlargement of mitochondria and intramitochondrial vesicles
formation.**

l-r) TEM images of mitochondria from wild-type, *rab32^l/rab32^l*,
Mef2>Dosmit, Mef2>Dosmit+Rab32^{WT}, Mef2>Dosmit+Rab32^{CA},
Mef2>Dosmit+Rab32^{DN}, and *rab32^l/rab32^l*; Mef2>Dosmit muscle tissue.
Scale bars: 2 μm. s) Number of vesicles per unit area for one-week-old flies
from panels l-r. n=80-85 from 2-3 flies. Statistical test: Mann-Whitney *U*
test.

Fig Reviewer#1 Q4-2 / Fig 8 c, e, f

The mitochondria associated protein, Rab32, colocalized with Tom20-HA and undergoes a protein–protein interaction with Dosmit.

a, b) Confocal microscopy images of YFP-Rab32 localized around mitochondria and colocalized with Dosmit in fly muscle tissue. Scale bars: 10 μm.

e) Western blotting of S2 cells transfected with HA-tagged Rab32. Endogenous Dosmit was immunoprecipitated using Dosmit antibody.

Rab32-HA was detected in the same complex. **f)** Western blotting of lysate co-transfected with Dosmit-HA and Rab32-eGFP, immunoprecipitated with GFP antibody. Dosmit-HA was detected in the same complex (starred).

Minor points:

Q5 1. The authors report that Dosmit and Rab32 can be detected in a mitochondria-enriched fraction (Figs. 2h, 5c and 8c). Since mitochondrial preparations are often contaminated with ER-localized proteins, ER marker proteins and an ER fraction should be shown as control.

A: Thank you for this comment; indeed, the ER marker KDEL could be detected in the mitochondrial fraction isolated using our methods for mitochondria purification (as shown below).

Q6 2. Do Dosmit and Rab32 localize to the mitochondrial outer membrane? This issue should be addressed by performing proteinase K treatment of intact and swollen mitochondria. Furthermore, do these proteins integrate into mitochondrial membranes?

A: Unfortunately, very few reports indicate that Proteinase K treatment is successful with *Drosophila* tissues ref. We tried for several months,

but were unable to obtain intact mitochondria after purification and
Proteinase K digestion. As an alternative, we used Tom20-HA as an
outer membrane marker, and we found that both Dosmit and Rab32
colocalized with Tom20-HA (Line # 193-197) (Fig Reviewer#1 Q6 /Sup
Fig 3, , shown below), indicating that Dosmit and Rab32 are probably
mitochondrial-outer-membrane proteins.

Fig Reviewer#1 Q6 /Sup 3

Dosmit was localized on the mitochondrial outer membrane.

a) Tom20-HA was ectopically expressed in muscle as a mitochondrial outer marker. Dosmit (Dsm) was colocalized with Tom20-HA. Scale bar: 10 μ m.

Q7 3. The authors should describe the screen in which Dosmit was originally identified. How many different genes were tested and were identified?

A: We screened 11 candidate mitochondrial genes, listed in the Materials and Methods section, for changes in mitochondrial morphology ((Line # 183-185)(Fig Reviewer #1 Q7 / now included as Supplementary Fig. 2, shown below), but knockdown of Tom20 and Tom40 led to embryonic death.

Fig Reviewer#1 Q7 / Sup Fig 2

Screening of mitochondrial morphology by knocking out nine mitochondria-associated proteins.

a) Mef2/+, MCU RNAi, MGE RNAi, Tom7 RNAi, Tom70 RNAi, Tomboy40 RNAi, ND39 RNAi, CG5662 RNAi, CG9393 RNAi, and Dosmit RNAi fly muscle stained with anti-ATP5A (green) and phalloidin (red). Scale bars: 10 μ m. **b)** Distribution of mitochondrial sizes for one-week-old Mef2/+, MCU RNAi, MGE RNAi, Tom7 RNAi, Tom70 RNAi, Tomboy40 RNAi, ND39 RNAi, CG5662 RNAi, CG9393 RNAi, and Dosmit RNAi flies. Significant differences are starred (Mann–Whitney *U* test; * $p < 0.05$; ** $p < 0.01$; *** $p < 0.001$; **** $p < 0.0001$). $n=80-100$ from 4-6 flies

**Q8 4. The authors used a *Dosmit* mutation (*Dosmit-EP*) to analyze**
**the role of the protein for mitochondrial morphology. The authors**
**should describe that mutant in more detail.**

296 A: *Dosmit-EP* is a mutated allele caused by a P-element insertion in the
297 27 base pair site of the *Dosmit* (aka *Cisd2*) coding region. This insertion
restricted translation of the functional protein. Further, we did not detect
any *Dosmit* protein in the *Dosmit-EP* mutants; more detailed information
is available at flybase *Dmel*\P{EP}*Cisd2*^{G6528}:

<http://flybase.org/reports/FBti0128420.html>

(Supplementary Fig. 4a, shown below).

**Fig Reviewer#1 Q8 / Sup Fig 4a**

P-element integrated into the first exon of *Dosmit-EP*.

Reviewer #2 (Remarks to the Author):

**1. Chen and colleagues describe the identification of Dosmit (also**
**known as Cisd2) in flies regulates mitochondrial morphology with**
**age and present data that suggests it may do so by promoting the**
**uptake of vesicles into mitochondria. This is a very interesting**
**premise as only a very few mechanisms of mitochondrial uptake**
**are known, and this would be a new one. As such, it is important to**
**ensure such claims are correct. The data presented include a**
**number of interesting observations, most notably, the presence of**
**multiple intramitochondrial vesicle-like structures and the**
**apparent uptake of cytosolic proteins, when Dosmit is**
**overexpressed. While data are presented that supports a**
**mechanistic role for Rab32 in this, it seems unclear exactly what**
**purpose this process may serve. One of the more striking**
**observations is that loss of Dosmit, via an EP element, and**
**presumably loss of vesicle uptake, appears to lead to a substantial**
**increase in lifespan and vice versa, although these data are**
**unfairly relegated to supplementary information. Is this the same**
**for 2-1 mutants? I'd like to know more about the organismal**
**phenotypes of Dosmit mutant and OE (or RNAi). Are they more**
**active into old age? Do they suppress markers of ageing?**

**Q1-1 A: In response to the question: “(...) loss of Dosmit, via an EP**
**element (...) appears to lead to a substantial increase in lifespan**
**(...). Is this the same for 2-1 mutants?”**

Yes, flies homozygous for Dosmit-2-1 exhibited phenotypes similar to
those observed for Dosmit-EP and RNAi flies (Supplementary Fig. 4c–
f). The smaller-mitochondria phenotype of Dosmit-EP and Dosmit-2-1
was fully rescued by the genomic clone (line#211-214) (Fig Reviewer
#2 Q1-1 /Sup Fig 5).

Fig Reviewer#2 Q1-1 /Sup Fig 5

The genomic sequence of Dosmit could suppress the circular mitochondrial phenotype of Dosmit-EP or Dosmit-2-1 flies.

a–g) Mitochondria from wild-type (**a, e**), Dosmit-EP (**b**), Dosmit/+; Dosmit-EP (**c**), Dosmit-2-1 (**f**), and Dosmit/+; Dosmit-2-1 (**g**) flies. The green channel indicates ATP5A staining and the red channel indicates phalloidin staining of F-actin in muscle. Scale bars: 10 μm . **d)** Quantification of mitochondrial sizes for wild-type, Dosmit-EP, and Dosmit/+; Dosmit-EP flies (mean \pm SD). Sample age: 7 d; n=80-100 from 8–10 flies. Statistical test: Mann–Whitney *U* test. **h)** Quantification of mitochondrial sizes for wild-type, Dosmit-2-1, and Dosmit/+; Dosmit-2-1 flies (mean \pm SD). Sample age: 7 d; n = 80-100 from 8–10 flies. Statistical test: Mann–Whitney *U* test. **i, j)** Western blot showing Dosmit re-expression in genetically rescued flies.

**Q1-2 In response to the question: “I’d like to know more about the**
**organismal phenotypes of Dosmit mutant and OE (or RNAi). Are**
**they more active into old age? Do they suppress markers of**
**ageing?”**

In previous *Drosophila* aging studies, two main organismal aging
makers have been used: locomotion and protein homeostasis. Protein
homeostasis is usually assessed by checking the accumulation of
ubiquitinated proteins in tissues such as muscle².
In the previous version of this paper, we compared Dosmit-EP with wild-
type flies when investigating longevity. In this revision, as per the
reviewer’s suggestion, we used both the Dosmit-EP and the Dosmit-2-1
mutants in the locomotion assay, and found that the climbing activity of
the mutants was maintained at a high level into middle age, while the
Dosmit-overexpressing, -EP and -2-1 lines exhibited much greater
decreases in climbing ability over the same period(line#271-277) (Fig
Reviewer #2 Q1-2-1 /Sup Fig 10).

. We also checked the other aging marker—protein ubiquitination—
in aged flies via immunostaining. Both the Dosmit-EP and Dosmit-2-1
flies had a greatly reduced accumulation of ubiquitinated protein
(Supplementary Fig. 12c–l, shown below). These results indicated that
the Dosmit mutants could suppress aging markers. In contrast, the
Dosmit-overexpressing flies had a severe accumulation of ubiquitinated
proteins in their mitochondria (line 423-436) (Fig Reviewer #2 Q1-2-2
404 /Sup Fig 12 shown below).

Fig Reviewer#2 Q1-2-1 / Sup Fig 10

In middle-age, Dosmit-mutant flies had a significantly greater climbing ability than wild-type flies.

Percentage of wild-type, Dosmit-overexpressing, Dosmit-EP, and Dosmit-2-1 flies categorized as exhibiting low, medium, and high activity. Dosmit-EP and Dosmit-2-1 flies showed significantly more climbing activity than wild-type flies in middle age (three weeks old). n = 100 flies. Statistical test: chi-square test.

Fig Reviewer#2 Q1-2-2 /Sup Fig 12

Ubiquitinated proteins accumulate within the mitochondria in ectopically expressed Dosmit flies and aged wild-type flies.

a, b) Mef2>mitoGFP (**a**) and Mef2>Dosmit+mitoGFP (**b**) fly muscle stained with ATP5A (green) and ubiquitin antibody (red). Ubiquitinated proteins accumulated in the mitochondria of Mef2>Dosmit+mitoGFP flies. **c, d)** Ubiquitinated proteins also accumulated in large puncta and were distributed throughout aged wild-type muscle, particularly in the intramitochondrial vesicles (arrowheads). **e–h)** Ubiquitinated proteins did not form or accumulate in either young or aged Dosmit-EP or Dosmit-2-1 flies. Scale bars: 10 μ m. **i, j)** Quantification of mitochondria in which ubiquitinated protein accumulated. n=80-100 from 4–6 flies. Statistical test: Mann–Whitney U test.

**Q2 Overall, this is an intriguing study but I'd like to know more**
**about these vesicles, e.g. see more examples of the double-**
**membrane bound vesicles (see more notes below). Determining**
**the 'content' of these vesicles is extremely important to**
**understanding this process. The cytosolic GFP experiment is a**
**good start but some results don't align well. For instance, the**
**fluorescence imaging, there appears to be strong re-localisation of**
**GFP coincident with the mitochondrial signal. One oddity is where**
**have all the myofibres gone? Is this representative?**

**Q2-1 A: In response to the question: "Determining the 'content' of**
**these vesicles is extremely important to understanding this**
**process."**

Determining the content of these vesicles requires time to resolve all the
details for each type of cellular trafficking vesicle. In this revision, we
have identified ubiquitinated proteins as one type of cargo of this type of
vesicle (please see Fig Reviewer #2 Q1-2-2 /Sup Fig 12 shown above).
One of the functions of intramitochondrial vesicles may thus be
transporting ubiquitinated proteins into mitochondria. Further LC/MASS
experiments may provide more details regarding the differences in
vesicle cargo between Dosmit-mutant and overexpressing lines in
future.

**Q2-2 In response to the question: "(...) some results don't align**
**well. For instance, the fluorescence imaging, there appears to be**
**strong re-localisation of GFP coincident with the mitochondrial**
**signal."**

To obtain more detailed imaging that would enable us to separate the
GFP and mitochondrial signals, we obtained confocal images of adult
muscle mitochondria, using a Zeiss LSM880 microscope (Objective LD
LCI Plan-Apochromat 40x/NA1.2), and processed them using ImageJ
software to construct a three-dimensional image and video(line 321-
326) (Fig Reviewer #2 Q2-2/Fig. 5g, also Sup fig 11 shown below). In
this imagery we could clearly see the specific GFP within the
mitochondria.

**Q2-3 In response to the question: "One oddity is where have all the**
**myofibres gone? Is this representative?"**

The muscle tissue under Dosmit overexpression had a protein
homeostasis defect, and we also observed fused and fewer muscle
fibers. Our observation of reduced locomotion supports this point.

Yes, the myofibers will indeed be lost under overexpression of Dosmit.

Fig Reviewer#2 Q2-2 /Sup Fig 5g

g) Three-dimensional imaging of Mef2>Dosmit+GFP fly mitochondria, with cytosolic GFP internalizing in the mitochondria.

**Q3** *It is already slightly odd that the ‘cytosolic’ GFP seems to*
*colocalise with the myofibres in the control condition, but I*
*appreciate that there is hardly any ‘cytosol’ in this tissue. More*
*importantly, the fractionation assay in Fig. 5c shows only a small*
*proportion of the GFP re-localises into the mitochondrial fraction.*
*This also doesn’t seem to reflect the immuno-EM quantification,*
*since this argues that ~70% of the molecules are in the*
*mitochondrial vesicles. It would also be useful to have a sense of*
*how well the immuno-EM detects the cytosolic GFP since hardly*
*any is visible in these images so it is hard to know how*
*representative the quantification in e is. Here, it would be useful to*
*know the relative distribution of GFP-gold in control samples.*
*Currently, it isn’t clear how much of the cytosolic GFP re-localises.*
*It should also be remembered that this is under transgenic*
*overexpression. Also, that co-localisation of sub-cellular*
*structures by light microscopy is difficult. What is the optical*
*thickness of the confocal sections taken here?*

496 A: In Dosmit–GFP-co-overexpressed flies we found that the majority of
497 cytosolic GFP localized in muscle fibers, with only small quantities
identified as being within mitochondria by immunostaining and EM. We
quantified the intensity of GFP the signal and found that the signal in the
mitochondrial fraction was ~15% of that of cytosolic GFP(Line335-337)
(Fig Reviewer #2 Q3/Fig. 5c, d, shown below), demonstrating the
internalization rate of Dosmit-driven mitochondria.

Another quantification showed that ~70% of gold particles were
localized inside intramitochondrial vesicles (Fig. 5f), which means that
~70% of intramitochondrial GFP particles are located within these
vesicles. To investigate this intra-vesicle GFP we focused on the
mitochondria (Fig. 5e), but most cytosolic GFP is localized in muscle
fibers.

Additionally, we created a 3D video of mitochondria in which Dosmit
and cytosolic GFP were ectopically coexpressed. In this video, cytosolic
GFP indeed moved into the enlarged mitochondrion (Fig. 5g, shown
same as above Fig Reviewer #2 Q2-2/Fig. 5g, and Supplementary Fig.
11).

Throughout the study we used a section thickness of 0.772 μm .

Fig Reviewer#2 Q3 / Fig 5c, d

c) Western blot showing GFP in the mitochondrial fraction of flight-muscle homogenates from flies ectopically expressing Dosmit (Mef2>GFP+Dosmit), but not in the mitochondrial fractions of flight-muscle expressing GFP without ectopic expression of Dosmit (Mef2>GFP) or control flies (wild-type). **d)** Quantification of band intensity showed that approximately 15% of the cytosolic GFP was internalized in mitochondria when Dosmit (Dsm) was ectopically expressed.

**Q4** *Another very important set of observations is the suppression*
*of age-related vesicles by Dosmit knockdown or mutation (Fig. 7),*
*but why was the RNAi used for one assay and the EP mutant for*
*another. It would be good to verify and extend these observations*
*by repeating the assays with multiple mutant conditions.*

525 A: We have now provided Dosmit-EP data, in Fig. 7a. We have also
repeated some experiments using Dosmit-2-1 mutants and have
included these data in Supplementary Figs 10 and 12 (shown above,
Fig Reviewer #2 Q1-2-1 /Sup Fig 10 and Fig Reviewer #2 Q1-2-2 /Sup
Fig 12). These results show that Dosmit-2-1 has a similar phenotype to
Dosmit-EP, including greater climbing activity (line 271-277)(Fig
Reviewer #2 Q1-2-1 /Sup Fig 10, no accumulation of ubiquitinated
proteins (Fig Reviewer #2 Q1-2-2 /Sup Fig 12), and maintenance of
small mitochondria during aging (Fig. 3e,f; Supplementary Fig. 12e–h).

**Q5** *The authors argue that the intramitochondrial vesicles are*
*double-membrane bound, possibly having originated as*
*invagination of the outer mitochondrial membrane. The cartoon in*
*Fig. 6h suggests that the authors envisage that the vesicles also*
*include inner membrane; (i) this idea can and should be tested*
*using an IMM marker (ATP5A should do), and (ii) this would imply*
*that they are triple-membrane bound. Is this observed?*

545 A: (i) We suspect that since mitochondria contain a high density of
546 inner-membrane proteins, only external outer membrane proteins would
be distinguishable from the background. We predict that gold particles
would not be distributed throughout the whole mitochondrial inner
membrane if inner membrane antibody (ATP5A) was used in immune-
EM.

Furthermore, the MOM is very close to the MIM, so the resolution of
the membrane structure in immune-EM is limited. There is therefore a
high possibility that we would not be able to distinguish whether
particles were localized to the MOM or the MIM in any case.

(ii) This is an interesting suggestion, but we have not observed any
triple-membrane-bound vesicles.

**Q6-1** *It would be appropriate to know more about the potential*
*mitochondrial localisation of Dosmit. The authors show*
*mitochondrial enrichment in a rude fractionation but a greater*
*depth is warranted especially if this mechanistically implicates*
*cooperation with Rab32 outside mitochondria. I.e. OMM*

**localisation could be shown by protease/digitonin assays,**
**carbonate extraction would determine the relative intergration etc.**
**For Fig. 8 d-g, I would appreciate seeing the split channels of the**
**high mag images to better appreciate the distribution of Rab32 and**
**Dosmit in the different conditions.**

571 A: One possible mitochondrial localization marker is Proteinase K.
However, very few reports indicate that Proteinase K treatment is
successful with *Drosophila* tissues ref. We tried for several months, but
were unable to obtain intact mitochondria after purification and
Proteinase K digestion. As an alternative, we used Tom20-HA as an
outer membrane marker, and found that both Dosmit and Rab32
colocalized with Tom20-HA (line 451-455) (see below Fig Reviewer #2
Q6-1 /Supplementary Fig. 3), indicating that Dosmit are probably
mitochondrial-outer-membrane proteins.

Fig Reviewer#2 Q6-1 /Sup 3

Dosmit was localized on the mitochondrial outer membrane.

a) Tom20-HA was ectopically expressed in muscle as a mitochondrial outer marker. Dosmit (Dsm) was colocalized with Tom20-HA. Scale bar: 10 μ m.

**Q6-2 In response to the question: For Fig. 8 d-g, I would appreciate**
**seeing the split channels of the high mag images to better**
**appreciate the distribution of Rab32 and Dosmit in the different**
**conditions.**

We have now added the image split into the different channels in Fig.
8d-g. See below

**Q7 Overall, the writing is reasonable but could do with a thorough**
**review to improve the flow and accuracy of the text and data, and**
**the current Abstract and Discussion are rather disjointed and**
**rambling.**

We have edited this revised version, and have modified the abstract
and discussion.

Minor:

**Q8 In general, the labels on the immunofluorescence micrographs**
**are hard to read and should be changed. Also, the indexing to the**
**respective data figures could be clearer. And the order of**
**Supplementary data seems to be quite random, e.g. Supp 4 follows**
**Supp 5 in the text.**

Fig. 1c. We have now enlarged the labels and made the figures easier
to read. We have also corrected the order of the supplementary figures.

**Q9 It isn't clear if a HA knock-in was made for Fis1. The text**
**suggests so but there no schematic in Supp. Fig. 1, and the Fig. 2d**
**suggests that a Fis1 antibody was used (and not anti-HA) but none**
**is listed in Methods. There is also a lack of consistency in the**
**Supp figures whether the gene names are capitalised or not.**
**(Follow FlyBase nomenclature).**

648 A: We did indeed generate the Fis1 antibody within our lab. Information
regarding the Fis1 antibody has been added to the Materials and
Methods section (line 737-738). We have also ensured that gene
names follow a consistent nomenclature.

**Q10 Lines 186-7: "a Dosmit-EP mutant... suggesting that Dosmit**
**may use Cys-Cys-linked dimer formation". This needs more**
**explanation.**

657 A: We have now amended this statement.

**Q11 Fig. 3a-b. Is the quantification for total Dosmit or just one**
**species?**

662 A: We only quantified the levels of the dimeric form of Dosmit (Dosmit*).

**Q12 Supp Fig. 4. The results are clear but for completeness a**
**statistical analysis should be applied. Also, reporting lifespan**
**parameters usually use median rather than mean.**

668 A: We have now tested for differences between the groups using the median,
which is the number of days taken to reach 50% mortality. The median
longevity for da/+ = 41 d; da>Dosmit = 24 d; wild-type = 42 d; and Dosmit-EP
= 55 d. This indicates that the median longevity of Dosmit-driven flies was
reduced by 41%, and that of the Dosmit-EP line was increased by 31%,
relative to the control (Supplementary Fig. 4).

**Q13 Fig. 4a. The scale bars don't seem to correspond well between**
**high and low mag images, especially the larval samples. Please**
**check.**

680 A: Thank you for drawing our attention to this error, we have now
corrected this mistake.

**Q14 the data showing the presence of these vesicles in Atg8**
**knockdown should be shown and explained in details, e.g. which**
**gene (Atg8a or b) was targeted? This also leads to a potential**
**issue with the experiment since both genes would need to be**
**targeted in order to block autophagy.**

689 A: We only knocked down Atg8a (VDRC109654). However, we
confirmed, using a western blot analysis, that the level of lipidation
during autophagy decreased following this knockdown (Fig. a, below).

**Q15 Line 267: “electronically dense” would be better expressed as**
**‘electron-dense’.**

700 A: This has now been changed accordingly.

**Q16 Lines 267-8: I do not understand the sentence “we found a**
**reduction in electronically dense material within these vesicles**
**compared to control muscle tissue”. What are the authors**
**comparing exactly? There are no/few vesicles in the control tissue.**
**Are they trying to compare with cytosol?**

708 A: Electron-dense areas usually indicate the location of membrane
compartments. We observed that the content of intramitochondrial
vesicles had a lower electron density, similar to that of cytoplasm, than
membrane-bound organelles, meaning that these vesicles were unlikely
to be organelles. Therefore, we proposed that the vesicles may be
involved in engulfing cytosolic proteins.

**Q17 Line 315: “Small-membrane vesicles”. What is a ‘small-**
**membrane’? Do the authors simply mean small, membrane-bound**
**vesicles? If so, they could just say ‘small vesicles’ (which, by**
**definition, are membrane-bound).**

720 A: Thank you for pointing this out. We did mean small, membrane-
721 bound vesicles, and have now changed this to “small vesicles”.

**Q18 Line 391: “Rab32 was found in abundance in the**
**mitochondria”. This claim seems like an overstatement, and**
**should be revised, given the rather faint signal shown.**

727 A: Agreed; we have now modified this statement.

**Q19 Lines 388-9. The authors seem to suggest that they did a**
**screen of Rabs but don’t show this, which seems unnecessary**
**and/or incomplete.**

733 A: Based on X Ao et al.⁴, we examined the genetic interaction between
734 three rab proteins (rab7, rab9, rab32) and Dosmit (Fig. a–h below). Only
735 Rab32 was able to partially suppress the Dosmit-induced mitochondria
phenotype.

Fig Reviewer#2 Q19

The Rab protein Rab32 mediates the Dosmit-induced enlargement of mitochondria, not Rab7, 9.

a) Mitochondria from: Mef2/+ control flies (b), flies with ectopic expression of Dosmit, b-h) all under Dosmit was ectopically expressed with c) Mef2>Rab7^{CA} e) Mef2>Rab9^{CA} g) Mef2>Rab32^{CA}, d) Mef2>Rab7^{DN} f) Mef2>Rab9^{DN} h), Mef2> Rab32^{DN} flies, only Mef2>Dosmit+YFP-Rab32^{CA} flies in which both Dosmit and the constitutively active GTP-binding Rab32 mutant (Rab32^{CA}) were ectopically expressed enhance the enlargement phenotype (h), Mef2>Dosmit+YFP-Rab32^{DN} flies in which both Dosmit and the dominant negative form of Rab32 (Rab32^{DN}) were ectopically expressed. CA: constative active form, DN: Dominant Negative.

**Q20 Lines 408-9: What do the authors mean by “homozygous**
**knockdown mutants”. Do they mean simply ‘homozygous**
**mutants’? They should also standardise their genetic**
**nomenclature. Here description of genes/transgene/mutants**
**should be Rab32.**

752 A: We have now edited this section accordingly.

**Q21 Lines 428-39: This section is rather incongruous. First, it**
**disrupts the flow placed here, but moreover, it doesn’t specifically**
**address the idea that the authors claim. It is interesting to know**
**that Drp1 overexpression can reverse the Dosmit overexpression**
**mitochondrial enlargement but this in no way implicates**
**mitophagy in the process or phenotype.**

**Similarly, the next section of Dosmit structure-function disrupts**
**the flow and would be better placed in the initial description of**
**Dosmit overexpression phenotypes.**

765 A: We would prefer to retain this part of the text. This experiment was
766 conducted with the aim of understanding the genetic interaction
between Dosmit and these major fusion and fission proteins
(Supplementary Fig. 14). The results indicate that Marf does not play a
substantial role in this type of mitochondrial enlargement, but still has an
impact on mitochondrial fusion. Under Dosmit and Marf overexpression,
the giant mitochondria fused into ever-larger ones (Supplementary Fig.
14f, g). Drp1 played a more striking role, and coexpression of both
Dosmit and Drp1 dramatically reduced the enlargement phenotype. We
agree, however, that there is as yet no proof for the process of
mitophagy by Drp1, and have removed this statement from the
manuscript.

The structure and function analysis will link to the mammalian
counterpart experiment, which was suggested by the comments of
Reviewer 3, who wanted to know whether these phenotypes are only
observed in *Drosophila* or whether they could be cross-species
conserved.

We identified that the N-terminus and CDGSH domain could be
crucial for Dosmit function. In mammals there are two paralogs in
mouse and human, Cisd1 and Cisd2. Performing a domain swap to
make a fused Cisd1-2 protein resulted in a similar phenotype to that
associated with Dosmit. We have now created all of the associated
transgenic flies and conducted cell-culture experiments, and have
added these results to the paper (Supplementary Fig. 16, 17).

**Q22 Lines 451-454: “we found that the N-terminus of Dosmit was**
**also required for the enlarged-mitochondria phenotype**
**(Supplementary Fig. 7d–f), suggesting that iron–sulfur binding**
**may play an important role in regulating mitochondrial dynamics.”**
**The first part of this sentence does not justify the second part.**

798 A: The CDGSH domain functions in iron–sulfur binding during the
799 mitochondrial respiratory chain. In fact, tissue with ectopically
expressed truncated Dosmit with the CDGSH domain deleted had a
slightly lower abundance of mitochondria (Supplementary Fig. 15).
Therefore, both Dosmit’s N-terminus and its CDGSH domain play an
important role in regulating mitochondrial morphology.

**Q23 Line 455-6: “our data suggests that Dosmit-induced transport**
**of vesicles into mitochondria which is dependent upon both Rab32**
**and Drp-1”. They haven’t shown that vesicle transport is**
**dependent on Drp1, only the enlargement phenotype.**

810 A: We have now edited this section accordingly.

**Q24 Lines 465-467: “Using an RNAi screen, we showed that the**
**major mitochondrial dynamic components Opa1, Drp1, Marf, and**
**Fis1 are not involved in vesicle formation”. This was not shown,**
**only mentioned in the text.**

817 A: We have removed this statement from the text.

**Q25 Lines 479-80: “are involved in subcellular localization”. What**
**does this mean?**

824 A: We found that although Dosmit mainly localizes on mitochondria in
*Drosophila melanogaster*, its functions are divided into Cisd1 and Cisd2
in mammals Hela cell. Cisd1 localizes on mitochondria, but Cisd2 is
found in the ER. The two Cisd proteins have different roles and
subcellular localization.

**Q26 Line 498: “(MDVs) have been reported to shuttle proteins**
**between the cytosol and mitochondria”. Not exactly as stated.**
**They have been shown to transport mitochondrial components to**
**other parts of the cell. This statement implies a reverse transport**
**direction which is not currently claimed.**

837 A: We have now edited this section as follows: "(MDVs) having been
reported to selectively import mitochondrial proteins and ultimately fuse
with peroxisomes."

Reviewer #3 (Remarks to the Author):

*In this paper, Chen et al. describes the phenotypes of Dosmit (aka*
*cisd2) expression and knockout in Drosophila. Dosmit is an iron-*
*sulfur binding protein and was identified to reduce the age-related*
*size increase of mitochondria when knocked down. Conversely,*
*upon expression of Dosmit, mitochondria contained double-*
*membraned vesicles that appeared to transport cytosolic cargo*
*and were positive for the outer mitochondrial membrane protein*
*TOM20. The mitochondrial localization of Dosmit is dependent on*
*Rab32 and Rab32 knock-out also leads to smaller mitochondria*
*similar to Dosmit knockout. Protein levels of Dosmit and*
*intramitochondrial vesicle formation increase with age of the flies.*
*The authors conclude that Dosmit induced transport of vesicles*
*might represent a therapeutically interesting novel route for*
*mitochondrial protein trafficking and that this pathway might*
*explain age-related mitochondrial changes.*

*1. While the appearance of double-membraned vesicles importing*
*cytosolic components into mitochondria is very interesting and*
*certainly novel, the manuscript seems largely descriptive. It is*
*unclear how Dosmit and Rab32 lead to the formation of*
*intramitochondrial vesicles, what their functions are, if any, and*
*what functional consequences the vesicles have. In addition, it is*
*unclear if the presence of double membrane intramitochondrial*
*vesicles is fly specific or a universal mechanism that also applies*
*to higher organisms. The authors point out that Dosmit knockout*
*has a beneficial effect on mitochondrial morphology and longevity*
*in Drosophila, but in mouse models cisd2 knockout shows quite*
*the opposite effect and recessive mutations in this gene are*
*associated with a severe neurodegenerative disease (Wolfram*
*syndrome) in humans. This leads to the question if the function of*
*Dosmit is conserved in higher organisms. If not this narrows the*
*scope of the presented work and it might be suited better for a*
*more specialized audience. It is ultimately up to the editor to*
*decide whether publication in Nature communications is justified.*

**Q1:A: In response to the question: “It is unclear how Dosmit and**
**Rab32 lead to the formation of intramitochondrial vesicles.”**

In the original version of this paper, we demonstrated a genetic
interaction between Dosmit and Rab32: while gain-of-function mutation
of the constitutively active form of Rab32 GTPase enhanced
mitochondrial enlargement, loss-of-function mutation (whether a
dominant-negative GTPase mutant or a GTPase-null mutant) greatly

reduced the phenotype. We also showed that Rab32 is associated with
mitochondria and mutually dependent on Dosmit for maintenance.
To address the question of how Dosmit and Rab32 cooperate to
mediate the formation of intramitochondrial vesicles in more detail, we
performed TEM to examine the correlations in previous genetic assay
data in detail (line # 494-497)(Fig Reviewer #3 Q1-1 /Sup Fig. 10 l-s)
indicates that there is a strong correlation between the activity of Rab32
enzyme and Dosmit in connection with intramitochondrial vesicles.
We also conducted an immune pull-down assay to investigate whether
there is a physical interaction between these two proteins (# line456-
460,and 630-631, Fig Reviewer#3 Q1-2 /Fig. 8c,e,f; Sup Fig 3) confirms
that there is such an interaction. These lines of evidence provide the
insight that these proteins cooperate to bring about vesicle formation.
It has been reported that Sorting nexin 6 (SNX6) is one effector
target of the Rab32 GTPase in mammals. Which protein is the effector
of *Drosophila* Rab32 and how they function together with Dosmit and
the Rab32-SNX protein complex to induce membrane curving will be
investigated in the near future.

Fig Reviewer#3 Q1-1 /Sup Fig 10 l-s

The mitochondria-associated protein Rab32 mediates the Dosmit-induced enlargement of mitochondria and intramitochondrial vesicles formation.

l-r) TEM images of mitochondria from wild-type, *rab32^l/rab32^l*, Mef2>Dosmit, Mef2>Dosmit+Rab32^{WT}, Mef2>Dosmit+Rab32^{CA}, Mef2>Dosmit+Rab32^{DN}, and *rab32^l/rab32^l*; Mef2>Dosmit muscle tissue. Scale bars: 2 μ m. s) Number of vesicles per unit area for one-week-old flies from panels l-r. n=80-85 from 2-3 flies. Statistical test: Mann-Whitney *U* test.

Fig Reviewer#3 Q1-2 /Fig. 8c, e,f; Sup Fig 3

The mitochondria associated protein, Rab32, colocalized with Tom20-HA and undergoes a protein–protein interaction with Dosmit.

a, b) Confocal microscopy images of YFP-Rab32 and Dosmit colocalized with Tom20-HA in fly muscle tissue. Scale bars: 10 μ m.

e) Western blotting of S2 cells transfected with HA-tagged Rab32. Endogenous Dosmit was immunoprecipitated using Dosmit antibody.

Rab32-HA was detected in the same complex (starred). **f)** Western blotting of lysate co-transfected with Dosmit-HA and Rab32-eGFP, immunoprecipitated with GFP antibody. Dosmit-HA was detected in the same complex (starred).

Q2: In response to the question: “(...) what their functions are, if any, and what functional consequences the vesicles have?”

We found that ubiquitinated proteins were present within intramitochondrial vesicles in aged control and all Dosmit-driven mitochondria (line 423-436, Fig. Reviewer #3 Q2 /Sup 12), suggesting that these vesicles play a role in transporting ubiquitinated proteins. The levels of Dosmit and ubiquitinated proteins increase during aging, suggesting that the functional consequences of Dosmit-driven vesicles may be involved in protein homeostasis and mitochondrial dysfunction.

Fig Reviewer#3 Q2 /Sup Fig 12

Ubiquitinated proteins accumulate within the mitochondria in ectopically expressed Dosmit flies and aged wild-type flies.

a, b) Mef2>mitoGFP (**a**) and Mef2>Dosmit+mitoGFP (**b**) fly muscle stained with ATP5A (green) and ubiquitin antibody (red). Ubiquitinated proteins accumulated in the mitochondria of Mef2>Dosmit+mitoGFP flies. **c, d)** Ubiquitinated proteins also accumulated in large puncta and were distributed throughout aged wild-type muscle, particularly in the intramitochondrial vesicles (arrowheads). **e–h)** Ubiquitinated proteins did not form or accumulate in either young or aged Dosmit-EP or Dosmit-2-1 flies. Scale bars: 10 μm . **i, j)** Quantification of mitochondria in which ubiquitinated protein accumulated. n=80-100 from 4–6 flies. Statistical test: Mann–Whitney *U* test.

Q3: In response to the question: “it is unclear if the presence of double membrane intramitochondrial vesicles is fly specific or a universal mechanism that also applies to higher organisms.”

The double-membrane intramitochondrial vesicles are driven by the Dosmit protein, and *Drosophila* Dosmit also induces mitochondrial enlargement. We tested this protein in both flies and cells from higher organisms. Expressing it in fly S2 cells and human HeLa cells induced similar phenotypes (line 547-568, Fig Reviewer #3 Q3-1 /Sup 17): mitochondrial enlargement and clustering, similar to the phenotype observed in transgenic fly tissue. This suggests that there is evolutionary conservation among these species.

Dosmit, also known as Cisd, is split into two paralogs in mammals: Cisd1 and Cisd2. Cisd2 contains a longer N-terminal domain, like that of *Drosophila* Dosmit, and it locates mainly on ER¹. Although Cisd1 lacks this longer N-terminal domain, it is located on mitochondria, which depends on its TM domain (data not shown here).

In an attempt to create a version of Cisd that was as close as possible to Dosmit, we created a hybrid form of Cisd1 and Cisd2 in which the longer N-terminal domain of Cisd2 (base pairs 1–40) was fused with the rest of Cisd1 (base pairs 11–108) (Supplementary Fig. 16a). This hybrid version we named N-mCisd2-mCisd1. This protein resulted in the enlargement of mitochondria in Dosmit-mutant flies under transgenic ectopic expression ((line 547-568, Fig. Reviewer #3 Q3-2 /Sup 16f). It also induced mitochondrial enlargement in transfected S2 cells (Fig. Reviewer #3 Q3-1 /Sup 17a), and in human HeLa cells (Fig. Reviewer #3 Q3-1 /Sup 17b). This suggests that the Dosmit phenotype is probably not specific to *Drosophila* and that it is probably conserved among species, although in higher organisms the gene has been split into two different versions. Further detailed examination may be required to improve our understanding of this mechanism.

Fig Reviewer#3 Q3-1 / Sup Fig 17

A hybrid protein in which the N-terminus of mouse Cisd2 was combined with mouse Cisd1 was sufficient for mitochondrial enlargement, but not mCisd1-HA, mCisd2-HA.

(a, b) Dosmit-HA, mCisd1-HA, mCisd2-HA, and N-mCisd2-mCisd1 ectopically and respectively expressed in S2 **(a)** and HeLa **(b)** cells stained with MitoTracker (red) and HA (green). Scale bars: 10 μm . **(c, d)**

Quantification of mitochondrial size in S2 **(c)** and HeLa **(d)** cells transfected with Dosmit, mCisd1, mCisd2, or N-mCisd2-mCisd1. n=80-100 from 8-10 cells. Statistical test: chi-square test. Scale bars: 10 μm .

Fig Reviewer#3 Q3-2 / Sup Fig 16

A hybrid transgene in which the N-terminus of mouse Cisd2 was combined with mouse Cisd1 suppressed fragmented mitochondrial phenotypes in Dosmit-mutated flies.

a) Construct illustration of the Dosmit (Dsm), mCisd1, mCisd2, N-mCisd2-mCisd1 genes. **b, c)** Dosmit-2-1 flies have smaller mitochondria than wild-type flies. **d–f)** Ectopic expression of mCisd1 (**d**) and mCisd2 (**e**) did not rescue the Dsm-2-1 phenotype, but N-mCisd2-mCisd1 (**f**) did partially induce mitochondrial enlargement. Scale bars: 10 μm .

Major points:

Q4 • As stated above, a major issue is that this phenotype seems to be limited to *Drosophila*

A: Please see our answer for Q3 above to the question about whether the double-membrane intramitochondrial vesicles are fly-specific or a universal mechanism.

Q5• Can the dosmit knockout phenotype(s) be rescued by expression of human *cisd1* or 2?

A: To confirm whether the function of Dosmit is conserved in mouse Cisd, we ectopically expressed mouse Cisd1 (mCisd1), Cisd2 (mCisd2), or N-mCisd2-mCisd1 in flies with a Dosmit-mutated background (Fig Reviewer #3 Q5 also same as Q3 Q3-2 /Sup 16).

We found that neither mCisd1 nor mCisd2 suppressed the small-mitochondria phenotype observed in the Dosmit-2-1 mutant; however, N-mCisd2-mCisd1 did suppress this phenotype. In addition, ectopic expression of mCisd1 and mCisd2 in *Drosophila* S2 and HeLa (same as Fig Reviewer #3 Q3-1 /Sup 17 cells could not induce mitochondrial enlargement, although N-mCisd2-mCisd1 did reproduce this phenotype. This suggests that there is conserved function among these species, although it has been divided between two different versions of the protein in the higher organisms.

Fig Reviewer#3 Q5 / Sup Fig 16

A hybrid transgene in which the N-terminus of mouse Cisd2 was combined with mouse Cisd1 suppressed fragmented mitochondrial phenotypes in Dosmit-mutated flies.

a) Construct illustration of the Dosmit (Dsm), mCisd1, mCisd2, N-mCisd2-mCisd1 genes. **b, c)** Dosmit-2-1 flies have smaller mitochondria than wild-type flies. **d–f)** Ectopic expression of mCisd1 (**d**) and mCisd2 (**e**) did not rescue the Dsm-2-1 phenotype, but N-mCisd2-mCisd1 (**f**) did partially induce mitochondrial enlargement. Scale bars: 10 μ m.

Q6• Adequate description of mitochondrial morphology quantification is lacking. The authors should provide a detailed description for each of the assessed parameter and the samples size(s).

A: We quantified mitochondrial morphology by measuring the volume of more than 80 mitochondria from 3–10 flies, using ImageJ software. We have now included all sample sizes (number of mitochondria and flies) throughout the manuscript.

Q7• Figure 3: Does the increase of Dosmit reflect the “increase in mitochondrial size within muscles as the flies aged” (line 129) and do other mitochondrial proteins increase with age, too? A mitochondrial loading control (ATP5A or Porin) should be added to exclude this possibility.

A: We checked the protein expression levels of many mitochondrial proteins in young and old flies (line 234-236, Fig Reviewer #3 Q7 /Sup Fig 6). We found no significant differences in protein levels between the groups for every mitochondrial protein we checked, apart from Dosmit.

Fig Reviewer#3 Q7 / Sup Fig 6

Dosmit level increased during aging but not other mitochondrial proteins

a) Western blotting of one- and eight-week-old wild-type flies. **b)** Fold-change of Porin, ATP5A, NDUFS3, Chchd3, and Dosmit in three-day- and eight-week-old w1118 flies. Statistical differences between ages are starred (Mann-Whitney *U* test: ** $p < 0.01$).

**Q8• *Supplementary Fig.4: Why is the analysis of longevity only***
***performed in male flies? Is there a statistical assessment?***

1070 A: We performed longevity experiments in male flies based on previous
publications^{2, 3}. Lifespan assays using males only are clearer. We have
now included a statistical assessment of survival rate.

**Q9• *Figure 6: the depicted model for vesicle formation shows that***
***the inner mitochondrial membrane forms the outer membrane of***
***the vesicles. The authors should support this model***
***experimentally by co-localizing inner mitochondrial markers to the***
***vesicles.***

1080 A: Our results show that while the enlarged mitochondria could be
stained with ATP5A (Fig. 4a), the vesicles could not. We predict that the
gold particles would be distributed throughout the whole mitochondrial
inner membrane (MIM) if the ATP5A antibody was applied via immune-
EM. In addition, since the mitochondrial outer membrane (MOM) is very
close to the MIM, and the resolution of the membrane structure in
immune-EM is limited, there is a strong possibility that we would not be
able to distinguish whether particles were localized to the MOM or the
MIM.

**Q10• *The authors mention that a small scale RNA screen led to the***
***identification of Dosmit and disqualified other candidates but the***
***data is not shown.***

1095 A: We screened 11 candidate mitochondrial genes, which are listed in
the Materials and Methods section, for changes in mitochondrial
morphology (line 183-185, Fig Reviewer #3 Q10/Sup Fig 2)., although
complete knockdown of Tom20 and Tom40 led to embryonic death.

Fig Reviewer#3 Q10 / Sup Fig 2

Screening of mitochondrial morphology by knocking out nine mitochondria-associated proteins.

a) Wild-type, MCU RNAi, MGE RNAi, Tom7 RNAi, Tom70 RNAi, Tomboy40 RNAi, ND39 RNAi, CG5662 RNAi, CG9393 RNAi, and Dosmit RNAi fly muscle stained with anti-ATP5A (green) and phalloidin (red). Scale bars: 10 μ m. **b)** Distribution of mitochondrial sizes for one-week-old w1118, MCU RNAi, MGE RNAi, Tom7 RNAi, Tom70 RNAi, Tomboy40 RNAi, ND39 RNAi, CG5662 RNAi, CG9393 RNAi, and Dosmit RNAi flies. Significant differences are starred (Mann–Whitney *U* test; * $p < 0.05$; ** $p < 0.01$; *** $p < 0.001$; **** $p < 0.0001$). $n=80-100$ from 4-6 flies

**Q11 • Line 282: The authors state that the uptake of cytosolic**
**components constitutes a novel route of communication between**
**the cytosol and the mitochondria, but it does not seem that**
**there is any specificity for the uptake nor that there is a release of**
**the vesicular components. As such, it is not clear how this**
**communication works and what it controls.**

1131 A: We cannot currently make conclusive statements about the
1132 specificity of the vesicles, but we can say that they are not completely
unspecific. We found that the cytosolic ribosome subunit Rps2 proteins
could be delivered into mitochondria, but Ref(2)p and Atg8a
accumulated next to mitochondria, but not inside vesicles (Fig Reviewer
3 #Q11 / not shown in text). This suggests that there might be selection
rules for cargo uptake.

We do not have enough data to prove whether the vesicular cargo
is released into the mitochondria, but an immunogold EM assay showed
that only a low percentage of GFP gold particles was localized inside
mitochondria but not inside vesicles (Fig. 5e). This indicates that the
cargo of intramitochondrial vesicles might be released into the
mitochondria. However, we need more evidence before we can be
certain about this.

The exact processes remain unclear, but we here provide a novel
perspective on mitochondrial transport and have identified the relevant
components (Dosmit/Rab32). We also speculate that Dosmit is a
possible mechanism of aging. Understanding the communication
between cytoplasm and mitochondria is indeed of the utmost
importance, but will require further research.

Fig Reviewer#3 Q11

Cytosolic ribosome subunit Rps2 proteins could be delivered into mitochondria, but not Ref(2)p and Atg8a

a) Mef2> Rps2-HA b) Mef2> Dosmit+Rps2-HA, a, b) stained with HA and ATP5A antibody. c) Mef2/+ and d) Mef2> Dosmit muscle stained with ATP5A and Ref(2)p antibody, Ref(2)p signals enhanced when Dosmit highly expressed. e) Mef2> mecherry-Atg8a and f) Mef2> Dosmit, mecherry-Atg8a stained with ATP5A. Only ribosome subunit Rps2 proteins could be delivered into mitochondria. Scale bar: 10µm.

Q12 • Line 388: “ Rab protein screening (using Rab32 as a target)”.
Please explain.

A: Our original hypothesis concerned the existence of regulators involved in the Dosmit-mediated pathway from cytoplasm to mitochondria. Therefore, we examined several Rab members reported in the literature⁴, to check if one has a genetic interaction with Dosmit via mitochondrial morphology. We identified Rab32 as the most likely candidate (Fig Reviewer#3 Q12).

Fig Reviewer#3 Q12**The Rab protein Rab32 mediates the Dosmit-induced enlargement of mitochondria, not Rab7, 9.**

a) Mitochondria from: Mef2/+ control flies (**b**), flies with ectopic expression of Dosmit, **b-h**) all under Dosmit was ectopically expressed with **c**) Mef2>Rab7^{CA} **e**) Mef2>Rab9^{CA} **g**) Mef2>Rab32^{CA} , **d**) Mef2>Rab7^{DN} **f**) Mef2>Rab9^{DN} **h**), Mef2> Rab32^{DN} flies, only Mef2>Dosmit+YFP-Rab32^{CA} flies in which both Dosmit and the constitutively active GTP-binding Rab32 mutant (Rab32^{CA}) were ectopically expressed enhance the enlargement phenotype (**h**), Mef2>Dosmit+YFP-Rab32^{DN} flies in which both Dosmit and the dominant negative form of Rab32 (Rab32^{DN}) were ectopically expressed. CA: constative active form, DN: Dominant Negative.

**Q13 • *The dependence of Dosmit and Rab32 on each other***
***regarding their mitochondrial localization is intriguing. It would be***
***good to strengthen the data using biochemical methods and***
***interesting to know if Dosmit and Rab32 (directly) interact with***
***each other.***

Please see our answer Reviewer#3 Q1-1, Q1-2 above to the question
about how Dosmit and Rab32 lead to the formation of intramitochondrial
vesicles.

**Q14 • *Figure 9: The authors state that the effect of Dosmit***
***expression was partially suppressed in flies coexpressing the***
***dominant negative Rba32 or in rab32 null mutant flies but this is***
***not really the case. To suppress the phenotype would mean that***
***the morphology should go back to normal however there are gross***
***abnormalities visible in Fig. 9i-k. How do the authors explain this?***

1196 A: The huge mitochondria could be not fully rescued to a nearly normal
phenotype by mutating Rab32 (Fig. 10j, k). We thus defined this
phenotype as “partially suppressed” and not “fully suppressed”. Also, in
this version we provide all of the EM data, which indicate that in
dominant-negative Rab32 or Rab32-null mutant flies, the number of
vesicles was indeed reduced, according to our statistical analysis (Fig.
10l-s).

Minor points:

**Q 15 •** In the abstract, line 23/24, the authors state that the Dosmit
protein forms vesicles. Please rephrase.

Yes, we have edited this state.

**Q 16•** Dosmit EP mutant is mentioned before it is introduced as
transposon mutant (line 186/187 and 189/190)

1214 A: Dosmit-EP is a transposon insertion mutant that disrupts the coding
sequence. The available information is in flybase FBst0030170. We
have now edited this section accordingly, and provide the construction
map of Dosmit-EP below (Supplementary Fig. 4a, shown below).

• **Line 391: “in abundance” should be deleted or a stron**

1223 A: We have now edited this section accordingly.

References

- 1. Wiley, S.E. *et al.* Wolfram Syndrome protein, Miner1, regulates sulphhydryl
 redox status, the unfolded protein response, and Ca²⁺ homeostasis. *EMBO*
 *Mol Med* **5**, 904-918 (2013).
- 2. Duncan, J.E., Lytle, N.K., Zuniga, A. & Goldstein, L.S. The Microtubule
 Regulatory Protein Stathmin Is Required to Maintain the Integrity of Axonal
 Microtubules in *Drosophila*. *PLoS One* **8**, e68324 (2013).
- 3. Teran, R. *et al.* The life span of *Drosophila melanogaster* is affected by
 melatonin and thioctic acid. *Invest Clin* **53**, 250-261 (2012).
- 4. Ao, X., Zou, L. & Wu, Y. Regulation of autophagy by the Rab GTPase network.
 *Cell Death Differ* **21**, 348-358 (2014).

Reviewers' comments:

Reviewer #1 (Remarks to the Author):

Chen and colleagues added a substantial amount of data in the revised version of their manuscript to support their conclusions. Several of my concerns have been addressed. The presented observations of a role of Dosmit in mitochondrial size and of intramitochondrial vesicles are interesting. My major concern is that the function of the intramitochondrial vesicles remains largely unclear. The authors report that these vesicles contain ubiquitinated proteins and propose a role of these vesicles in cellular protein homeostasis. However, additional data supporting this notion are missing. Overall, the reported findings are interesting, but remain largely descriptive.

Specific comments:

Figures 2h, 5c and 8h: The authors have to clearly indicate that the fraction M does not represent a mitochondrial fraction but rather a cellular membrane fraction containing microsomes. The authors should show in addition highly-purified mitochondria with reduced ER contamination to reveal the mitochondrial localization of Dosmit. Furthermore, it remains unclear whether Dosmit localizes on the mitochondrial surface or inside mitochondria.

Figure 8e and 8f: The authors should add appropriate negative controls to reveal the specificity of the presented immunoprecipitations. Further, they should indicate how much load and elution is shown to reveal whether this is a significant interaction or not.

Reviewer #2 (Remarks to the Author):

I appreciate that the authors have worked hard at addressing the majority of the reviewer queries and I am reasonably satisfied with the answers to my own question and criticisms, although there are still some points that were missed or inadequately answered that I detail below. Clearly, there are still many aspects of this intriguing set of observations to elaborate which I appreciate may be beyond the scope of this study, but there are still some important aspects raised between multiple reviewers that deserve addressing, such as demonstrating that GFP or Dosmit is truly internal to mitochondria.

Q3. Fig. 5d. Currently, the chart Y axis label doesn't make sense as it describes the amount of GFP in mitochondria, which for the control (GFP (C)) condition should be 0 not 100. So better expressed as % of total GFP signal (and the legend should make it clear that this is quantification of c).

It would be relevant to determine that the 'intra-mitochondrial' GFP in these fractions is indeed inside by applying a proteinase/digitonin analysis to the mitochondrial fractions. Internal GFP would initially be protected (whereas external 'sticky' GFP wouldn't) and eventually degraded upon permeabilisation.

[A side note: it is confusing that the authors refer to Fig. 5g simultaneously as Supp Fig. 11, but Supp Fig. 11 doesn't seem to exist.]

Q4. As I said before – it would be good to show results from multiple manipulations. I wasn't saying replace the KD with EP, so why not show the KD as well as the EP data.

Q5ii. If the authors haven't seen triple membraned vesicles, I think they should revise their model. It could simply be an invagination of the OMM encapsulating the IMM.

Q6i. I am confused by the statement "One possible mitochondrial localization marker is Proteinase K".

What does this mean? Proteinase K is not a marker for mitochondria. What I meant was not to perform proteinase/digitonin or carbonate extraction on tissues but on isolated mitochondria (see above for GFP localisation).

Q8. The authors didn't get my point – it is very hard to read green/red coloured writing on a green/red coloured immunofluorescence micrograph. These labels could easily be placed outside of the images (especially where they are the same labels between images e.g. Fig. 1 ATP5A/Phal (?) is common to all panel a, so could be outside of these images). The same applies throughout the figures.

Q12. I didn't raise this before but the increased longevity is a striking phenotype, as very few manipulations genuinely extend lifespan. But looking closer this result may be skewed by the control having an abnormal shortened lifespan particularly in early ages (10-40 days). True WT should mostly be alive at this stage, like the EP. Did the authors outcross either their 'WT' (da/+) or EP for several generations? If not, this should be stated in Methods.

Q14. This is a poor answer to this question: the authors clearly do not want to show the presence of vesicles upon Atg8a KD - why? I asked for the details of the Atg8 not for my benefit, but for the readers. It should be stated that they targeted Atg8a, and detail the line they used in the Methods (this hasn't been done). It would also be useful to show the GABARAP blot presented to the reviewers.

Q26. The response to this query is now also incorrect as it implies that MDVs have ONLY been implicated in traffic of mitochondrial vesicles to peroxisomes, which is not the case. I suggest the authors review the literature that they are citing and adjust their statements (and knowledge) accordingly.

Additionally; the authors discuss the homology of Dosmit to mammalian CISD1 and 2 but it would be useful to comment on the homology (or lack) with CISD3.

Reviewer #3 (Remarks to the Author):

The authors present an improved manuscript that addresses most of my comments and concerns adequately. The finding that ubiquitinated proteins are found in these vesicles is intriguing and it would be great to find out more about the biological role and downstream function of these vesicles. This reviewer acknowledges, however, that this might be beyond of the scope of the current manuscript.

Minor:

There is some sections that need to be proof-read/edited: 454-460, 566-568.

Line 555-556 is unclear: Do the authors want to state that fly Dosmit is localized to mitochondria when expressed in HeLa cells?

Here follows our point-by-point reply:

**Reviewers' comments:**

**Reviewer #1 (Remarks to the Author):**

**Chen and colleagues added a substantial amount of data in the revised version of**
**their manuscript to support their conclusions. Several of my concerns have been**
**addressed. The presented observations of a role of Dosmit in mitochondrial size**
**and of intramitochondrial vesicles are interesting. My major concern is that the**
**function of the intramitochondrial vesicles remains largely unclear.**

**The authors report that these vesicles contain ubiquitinated proteins and**
**propose a role of these vesicles in cellular protein homeostasis. However,**
**additional data supporting this notion are missing. Overall, the reported findings**
**are interesting, but remain largely descriptive.**

**the function of the intramitochondrial vesicles remains largely unclear.**

**These vesicles contain ubiquitinated proteins and propose a role of these vesicles**
**in cellular protein homeostasis. However, additional data supporting this notion**
**are missing.**

Reply: Mitochondria have previously been described as an organelle for degrading
unfolded proteins that, in a yeast model system, are delivered from the cytosol¹. This
group also conducted a genetic screen for 13 nonessential mitochondrial proteases,
processing peptidases, or oligopeptidases, determining the effects of their loss on the
dissolution of protein aggregates in mitochondria¹. Yeast PIM1, also known as
protease Lon, plays a major role in dissolving aggregates, and accordingly, the
deletion of PIM1(LON) had the strongest effect (Q1 PTP Fig. 1).

We wanted to test the hypothesis that *Drosophila* mitochondrial proteases play a
similar role in the degradation of ubiquitinated proteins within Dosmit-induced
vesicles and identify the potential function of these vesicles. We did a small-scale
RNAi screen of the mitochondrial proteases SPG7, Rhomboid-7, HtrA2, dj-1beta, and
Lon. Following the loss of the Lon protease (i.e., via Lon RNAi), *Drosophila*

exhibited a normal hatching rate (Supplementary Fig. 18f), similar to the Dosmit
overexpression line. However, we observed a synergistic lethal phenotype when Lon
RNAi was combined with high Dosmit expression (Supplementary Fig. 18f), which
suggested a genetic interaction between Lon and Dosmit. The Lon RNAi line
exhibited induced hyper-ubiquitination of proteins within mitochondria when Dosmit
was highly expressed (Supplementary Fig. 18d, e). This is in accordance with the
previous finding that the yeast Lon protease may be involved in mitochondrial protein
homeostasis.

Q1 PTP Fig 1: Loss of pim1 function led to the lowest efficiency in the degradation of misfolded proteins (red line)¹.

In this revision we further provide an additional line of evidence that Lon protease
may be involved in protein homeostasis for ubiquitinated protein degradation in these
vesicles. This new data has been added to the text in lines 471–477:

Supplementary Figure 18. Ubiquitinated proteins accumulate in mitochondria when Lon protease is knocked down. (a, b) Knocking down Lon protease leads to a widespread increase in ubiquitinated proteins. (c) Ectopic Dosmit expression causes ubiquitinated protein accumulation in mitochondria. (d) Knocking down Lon protease with Dosmit overexpression caused a greater accumulation of ubiquitinated protein than Dosmit overexpression alone. Scale bar: 5 μ m. Level of (e) ubiquitin accumulation ($n = 3$) and (f) hatch rate ($n = 100$ flies). Statistical test: Student's t test. Source data are provided as a Source Data file.

**Specific comments:**

**Figures 2h, 5c and 8h: The authors have to clearly**
**indicate that the fraction M does not represent a mitochondrial fraction but**
**rather a cellular membrane fraction containing microsomes. The authors should**
**show in addition highly-purified mitochondria. with reduced ER contamination**
**to reveal the mitochondrial localization of Dosmit. Furthermore, it remains**
**unclear whether Dosmit localizes on the mitochondrial surface or inside**
**mitochondria.**

Reply: We agree with the reviewer's comment that the M fraction may have
contained other cellular membranes (such as ER). We have modified the statement in
the Materials and Methods section, lines 884–885.

In order to assess the localization of Dosmit in the mitochondria with greater
accuracy, we used transgenic fly lines with KDEL-fused GFP as ER marker, and
Golgi-targeting peptide from B4GALT1 fused with GFP, this Golgi-GFP as Golgi
marker (add into material and method line (753-755)). Muscle tissue from each of
these transgenic fly lines was co-immunostained with Dosmit and ATP5A, an inner
membrane protein. As shown in Supplementary Fig. 19, most of the Dosmit protein
was colocalized with the mitochondria. In particular, most of the Dosmit signal (red)
was located on the edge of the mitochondria, with 99.48% colocalized with ATP5A,
0.52% colocalized with the ER marker, and 0.07% with the Golgi marker. This
suggests that Dosmit mainly colocalizes with mitochondria but not with other
membrane-bound organelles. We have added this new data to the text in lines 197-
202.

Supplementary Figure 19. ER and Golgi markers are not colocalized with Dosmit in muscle tissue. Dosmit is mainly expressed in mitochondria and does not colocalize with (a) KDEL-GFP or (b) Golgi-GFP (n = 80 from 2–3 flies). Scale bar: 5 μ m. (c) Quantification of the percentage which Dosmit colocalized with ATP5A, ER-GFP, Golgi-GFP. Statistical test: Mann–Whitney U test. Source data are provided as a Source Data file.

We then wanted to assess the exact localization of Dosmit in the mitochondria.
Reviewers 1 and 2 both suggested using the digitonin method to investigate this
question.
We added a small quantity of digitonin (40 μm) to S2 cell cultures permeabilized via
proteinase K digestion. Dosmit was promptly degraded after 15 min of 50 ng/ μl
protease digestion following the mild digitonin (40 μm) treatment. In comparison, the
outer membrane protein Porin was degraded after 60 min of treatment, suggesting that
Dosmit is relatively easy to access and degrade under mild cell permeabilization
conditions. For the mitochondrial inner membrane protein ATP5A (an indicator of
proteins inside the mitochondria), protein levels remained stable in low digitonin
conditions (Supplementary Fig. 20a), but all mitochondrial proteins could be degraded
with a high concentration of digitonin (1600 μm), which permeates the mitochondrial
membrane (Supplementary Fig. 20b).

Supplementary Figure 20. Dosmit protein and an outer mitochondrial protein, Porin, are simultaneously digested with proteinase K for 60 min in digitonin-treated S2 cells. (a) In S2 cells treated with a low titer of digitonin, Dosmit and Porin proteins are completely digested when treated with proteinase K (PK) for 60 min. (b) Porin and an inner membrane protein (ATP5A) are completely digested in 15 min with a higher titer of digitonin. Source data are provided as a Source Data file.

Furthermore, we wished to compare these results via immunostaining performed
under the same conditions (Supplementary Fig. 21). After undergoing the treatment
above, S2 cells were stained with antibodies against the inner membrane protein
ATP5A (green) and Dosmit (red), as well as with DAPI (blue). The Dosmit signal
was located on the edge of the mitochondria, very similar to our *in vivo* data (Sup.
Fig. 19). A small quantity of digitonin (40 μm) was used to treat the cells. After 15
118 min of protease digestion, the Dosmit signal was greatly diminished (< 20% of
119 original levels), and after 30 min the Dosmit signal had disappeared (Sup. Fig. 21).
These lines of evidence suggest that Dosmit is located on the surface of mitochondria.
We have added this new data to the text in lines 203-218.

Supplementary Figure 21. Dosmit protein is digested faster than ATP5A when S2 cells are treated with digitonin/proteinase K. (a) An S2 cell without digitonin/proteinase K (PK) treatment. (b) An S2 cell treated with PK only for 15 min. (c–e) S2 cells treated with digitonin for the same length of time, but with PK for 15, 30, or 60 min. Scale bar: 5 μm . (f) Percentage of Dosmit-associated mitochondria in each treatment (n = 70-80 from 4 cells). Statistical test: Mann–Whitney U test. Source data are provided as a Source Data file.

**Figure 8e and 8f: The authors should add appropriate negative controls to reveal**
 **the specificity of the presented immunoprecipitations. Further, they should**
 **indicate how much load and elution is shown to reveal whether this is a**
 **significant interaction or not.**

Reply: We thank the reviewer for this suggestion. We used mitochondrial matrix
 protein Hsc-70-5 as a negative control (Fig. 8). Hsc-70-5 3XHA is described in the
 Materials and Methods section in lines 777-783.

The pull-down assay indicates a specific interaction between Rab32 and Dosmit. Each
 input was loaded with 10 µg of total protein extract. The eluted protein came from an
 immunoprecipitation of 500 µg of cell extract.

We have used this data in the new Fig 8. (Text in line 497-502)

New Fig 8. Dosmit and Rab32 show protein–protein interactions in S2 cells. S2 cells transfected with Hsc70-5-HA or Rab32-HA plasmids. (a) When Rab32-HA or Hsc70-5-HA is immunoprecipitated with an anti-HA antibody, Dosmit can be detected in the Rab32-bound complex. (b) When endogenous Dosmit proteins are immunoprecipitated with the anti-Dosmit antibody, only Rab32-HA can be detected in the Dosmit–pulled-down complex. For the pull-down, 10 ug of input was loaded. For SDS-PAGE, 500 ug protein from the immunoprecipitation was loaded. Source data are provided as a Source Data file.

**Reviewer #2 (Remarks to the Author):**

**I appreciate that the authors have worked hard at addressing the majority of the**
**reviewer queries and I am reasonably satisfied with the answers to my own**
**question and criticisms, although there are still some point that were missed or**
**inadequately answered that I detail below. Clearly, there are still many aspects**
**of this intriguing set of observations to elaborate which I appreciate may be**
**beyond the scope of this study, but there are still some important aspects raised**
**between multiple reviewers that deserve addressing, such as demonstrating that**
**GFP or Dosmit is truly internal to mitochondria.**

**Q3. Fig. 5d. Currently, the chart Y axis label doesn't make sense as it describes**
**the amount of GFP in mitochondria, which for the control (GFP (C)) condition**
**should be 0 not 100. So better expressed as % of total GFP signal (and the legend**
**should make it clear that this is quantification of c).**

Reply: Thank you for pointing out this error. We have corrected the *y*-axis label in

Fig. 5d to read “% of GFP signal”, as per your suggestion.

**It would be relevant to determine that the ‘intra-mitochondrial’ GFP in these**
**fractions is indeed inside by applying a proteinase/digitonin analysis to the**
**mitochondrial fractions. Internal GFP would initially be protected (whereas**
**external ‘sticky’ GFP wouldn’t) and eventually degraded upon permeabilisation.**

Reply: To address this question, we used digitonin and proteinase K to treat the
muscles of adults in which GFP was coexpressed with Dosmit. After proteinase K
digestion, GFP was localized only in the mitochondria as numerous puncta
(Supplementary Fig. 22b), while cytosolic GFP was mostly degraded (Supplementary
Fig. 22a, b). We found an equivalent result when using Western blotting to investigate
GFP in the mitochondrial fraction of flight-muscle homogenates under digitonin and
proteinase K treatment from flies ectopically expressing Dosmit
(Mef2>GFP+Dosmit); the cytosolic fraction was mostly degraded, but the
mitochondrial fraction remained at the same level. (Supplementary Fig. 22c)

We have now edited in the main text (lines 362–371) and Materials and Methods
section (lines 844–853) appropriately.

Supplementary Figure 22. Cytosolic GFP remains within mitochondria after digitonin/proteinase K treatment.

(a) Mef2>Dosmit+GFP muscle not treated with digitonin and proteinase K (PK). (b) Cytosolic GFP was digested when treated with digitonin and proteinase K. Scale bar: 2.5 μm. (c) Western blot showing GFP in the mitochondrial fraction of flight-muscle homogenates under digitonin and proteinase K treatment from flies ectopically expressing Dosmit (Mef2>GFP+Dosmit), the cytosolic fraction was mostly degraded, but mitochondrial fraction remained the same level. Source data are provided as a Source Data file.

**[A side note: it is confusing that the authors refer to Fig. 5g simultaneously as**
**Supp Fig. 11, but Supp Fig. 11 doesn't seem to exist.]**

Reply: Thank you for bringing this error to our attention; it should have been

“Supplementary Data 11/ Movie 1”. We have corrected this reference to “(Fig. 5g;

Supplementary Data 11/ Movie 1)” in line 351.

**Q4. As I said before – it would be good to show results from multiple**
**manipulations. I wasn't saying replace the KD with EP,**

Reply: We have added the EP mutant line and RNAi (KD) line data to Figure 7 for a

better comparison. The following figure replaces the original Figure 7:

**Q5ii. If the authors haven't seen triple membraned vesicles, I think they should**
**revise their model. It could simple be an invagination of the OMM encapsulating**
**the IMM.**

Reply: We have modified the model figure to avoid this potential misunderstanding.

In the new model, we have indicated the cytoplasm, with the green circles

representing GFP protein. The original figure has thus been replaced with the new

model figure, shown below.

**Q6i. I am confused by the statement “One possible mitochondrial localization**
**marker is Proteinase K”.** What does this mean? **Proteinase K is not a marker for**
**mitochondria. What I meant was not to perform proteinase/digitonin or**
**carbonate extraction on tissues but on isolated mitochondria (see above for GFP**
**localisation).**

Reply: We apologize for this error; we meant “one possible way to examine

mitochondrial localization is by using proteinase K digestion combined with digitonin

treatment.” Again, we thank the reviewer for their suggestion. In this revision, we

have completed these experiments and displayed the results in Supplementary Fig.

20–22, as shown in replies to other comments above.

**Q8. The authors didn't get my point – it is very hard to read green/red coloured**
**writing on a green/red coloured immunofluorescence micrograph. These labels**
**could easily be placed outside of the images (especially where they are the same**
**labels between images e.g. Fig. 1 ATP5A/Phal (?) is common to all panel a, so**
**could be outside of these images). The same applies throughout the figures.**

Reply: We have now removed all the coloured labels from the images and use only
one label for all similar images. Figs. 1, 2, 3, 4, 7, 8, 9, and 10, as well as
Supplementary Figs. 2, 4, 5, 8, 13, 14, and 15 have all been corrected.

**Q12. I didn't raise this before but the increased longevity is a striking phenotype,**
**as very few manipulations genuinely extend lifespan. But looking closer this**
**result may be skewed by the control having an abnormal shortened lifespan**
**particularly in early ages (10–40 days). True WT should mostly be alive at this**
**stage, like the EP. Did the authors outcross either their 'WT' (da/+) or EP for**
**several generations? If not, this should be stated in Methods.**

Reply: We backcrossed the EP mutant to wild-type w^{1118} for 10 generations, and the
CRISPR mutant for five generations (this information has been added to the Materials
and Methods section in lines 820-823). We repeated the lifespan experiments for
Dosmit EP and KO mutants as well as genomic rescue lines, with the new data
replacing the original Supplementary Fig. 9. These repeats did not change our
conclusions however and there was a slight lifespan extension for both the Dosmit EP
and the KO mutants (lines 287–294).

Supplementary Fig. 9. Effects of Dosmit overexpression and knock-out on the mean lifespan of *Drosophila melanogaster*. (a) Dosmit expression improved the mean lifespan of Dosmit-EP and Dosmit 2-1 flies ($n = 100$ flies). (b) Flies that ectopically expressed Dosmit in their muscles exhibited a shorter mean lifespan than Mef2/+ flies ($n = 100$ flies). The mean lifespans were as follows: wild-type, 62.6 d; Dosmit-EP, 73.3 d; Dosmit 2-1, 76 d; Dosmit/+; Dosmit-EP, 65.7 d; Dosmit/+; Dosmit 2-1, 64.7 d; Mef2/+, 62.3 d; and Mef2>Dosmit, 47.8 d. Source data are provided as a Source Data file.

**Q14. This is a poor answer to this question: the authors clearly do not want to**
**show the presence of vesicles upon Atg8a KD - why? I asked for the details of the**
**Atg8 not for my benefit, but for the readers. It should be stated that they**
**targeted Atg8a, and detail the line they used in the Methods (this hasn't been**
**done). It would also be useful to show the GABARAP blot presented to the**
**reviewers.**

Reply: We appreciate the reviewer's suggestion and concerns regarding whether
silencing Atg8a reduces vesicle formation. In *Drosophila*, there are two Atg8
isoforms: Atg8a and Atg8b. Atg8a is ubiquitously expressed, while Atg8b is
specifically expressed in testis (FlyBase RNAseq data). Atg8a is thought to play a role
in autophagy in most tissues, while Atg8b has more specific roles in the testis. In
addition, *Drosophila* Atg8a is recognized by the anti-human GABARAP antibody. In
the previous revision, we showed that RNAi against Atg8a inhibits the lipidation of
LC3. Here, RNAi knockdown of Atg8a did not affect vesicle formation, as revealed
by immunostaining and TEM examination (Sup. Fig. 23b, d). Intramitochondrial
vesicles were still formed without Atg8a, suggesting that these vesicles may not be
autophagosomes.

We have added this result in lines 320-324.

Supplementary Figure 23. Atg8a knocking down could not suppress formation of intra-mitochondrial vesicles. (a) Mitochondria morphology of Mef2/+, (b) Mef2>Atg8a RNAi, (c) Mef2>Dosmit, and (d) Mef2>Dosmit+Atg8a RNAi. Scale bar: 5 μ m (fluorescence image), 1 μ m (EM image). (e) Quantification of intra-mitochondrial number. N=4-5 from 2 flies. Statistical test: t-test. Source data are provided as a Source Data file.

**Q26. The response to this query is now also incorrect as it implies that MDVs**
**have ONLY been implicated in traffic of mitochondrial vesicles to peroxisomes,**
**which is not the case. I suggest the authors review the literature that they are**
**citing and adjust their statements (and knowledge) accordingly.**

Reply: We appreciate the reviewer’s comment. We have now corrected the statement
to: “MDVs have been reported to selectively import mitochondrial proteins and fuse
with inter organelles². MDVs are also important for mitochondrial quality control,
dynamics, and antigen presentation³. Recent reports suggest that MDVs deliver the
peroxide-generating enzyme Sod2 to bacteria-containing phagosomes⁴.” (lines 642-
646)

**Additionally; the authors discuss the homology of Dosmit to mammalian CISD1**
**and 2 but it would be useful to comment on the homology (or lack) with CISD3.**

Reply: The iron–sulphur (2Fe-2S) binding motif CDGSH proteins regulate iron and
reactive oxygen metabolism⁵. This protein family includes mammalian CISD1–3,
which are three proteins involved in diabetes, obesity, cancer, aging, cardiovascular
disease, and neurodegeneration⁵. In this study, we found that *Drosophila* has only two
CDGSH proteins: Dosmit (between CISD1 and CISD2) and CG3070 (CISD3). The
CDGSH domain appeared early in evolution, and CISD1 and -2 diverged about 650–
720 million years ago⁵. Both the human and fly CISD3 proteins contain two CDGSH
domains, although whether CISD3 plays similar roles to Dosmit requires further
study.

We have added a comment on CISD1, -2, and -3 in the Discussion section, in lines
724–732.

**Reviewer #3 (Remarks to the Author):**

**The authors present an improved manuscript that addresses most of my**
**comments and concerns adequately. The finding that ubiquitinated proteins are**
**found in these vesicles is intriguing and it would be great to find out more about**
**the biological role and downstream function of these vesicles. This reviewer**
**acknowledges, however, that this might be beyond of the scope of the current**
**manuscript.**

Reply: We thank the reviewer for this comment. We have now discovered more about
the biological role and function of these vesicles. Please find the answer included in
our reply to Reviewer 1's Question 1. However, we agree that further investigation
into the function of these vesicles is needed as part of future studies.

**Minor:**

**There is some sections that need to be proof-read/edited: 454-460, 566-568.**

**Line 555-556 is unclear: Do the authors want to state that fly Dosmit is localized**
**to mitochondria when expressed in HeLa cells?**

Reply: Thank you for this comment; the following paragraphs have been re-edited.

line454-460 (original text) → line 497-502 (re-edited)

When coexpressed, the mitochondrial outer membrane marker Tom20-HA colocalized
with YFP-Rab32 (YFP: yellow fluorescent protein) and Dosmit, which suggested they
were all probably located on the outer membrane (Fig. 8a, c; Supplementary Fig. 3).

Further, Rab32 did not simply share a location with Dosmit. A co-immune pull-down
assay in S2 cells revealed Dosmit protein in the Rab32-HA-contained
immunocomplex that was pulled down by the HA antibody (Fig. 8e), and Rab32-HA
protein but not Hsc70-5-HA in the Dosmit-associated complex that was pulled down
by the Dosmit antibody. This indicates that Dosmit and Rab32 form a protein complex
(Fig. 8f).

line 566-568 (original text) → line 580-603 (re-edited)

**Mouse cisd-domain hybrid protein may cause mitochondrial enlargement and**
**rescue *Drosophila* Dosmit mutants.**

To test whether the Dosmit-induced mitochondrial phenotype is conserved across
species, we identified and generated mouse cDNA clones for the mammalian homolog
of Dosmit, Cisd. This protein family contains two members, Cisd1 and Cisd2; Cisd2
has a longer N-terminal domain, similar to Dosmit (Supplementary Fig. 16a), but its
subcellular localization is in the ER, whereas Cisd1 is located mainly on mitochondria
(Supplementary Fig. 16d, e and 17a, b). *Drosophila* Dosmit thus apparently combines
the features of Cisd1 and Cisd2 (Supplementary Fig. 16a), in that it is expressed in fly
muscle, S2 cells, and HeLa cells, and it localizes on mitochondria (Supplementary
Fig. 17a).

To understand whether Dosmit and the Cisd family are functionally conserved
proteins, we created Dosmit-2-1 mutant cells that ectopically expressed mouse-Cisd1
(mCisd1), mouse-Cisd2 (mCisd2), or a hybrid protein that fused the long mouse- N-
terminal domain of Cisd2 (1-40) with Cisd1 (11-108)/ (N-mCisd2- mCisd1)
(Supplementary Fig. 16). We found that the rounded mitochondria phenotype was
only suppressed by expression of the N-mCisd2-mCisd1 hybrid protein, but not by
mCisd1, mCisd2 expression (Supplementary Fig. 16f), although some of the

mitochondria became larger than in wild-type cells (Supplementary Fig. 16). In
addition to this, N-mCisd2-mCisd1 expression resulted in mitochondrial enlargement
in both *Drosophila* S2 cells and human HeLa cells (Supplementary Fig. 17a–d).
In addition, N-mCisd2-mCisd1 resulted in mitochondrial enlargement in both
*Drosophila* S2 cells and human HeLa cells (Sup. Fig. 17a–d). These results suggest
that Dosmit is an earlier evolutionary form of the Cisd protein family.

**Line 555-556**

**whereas Cisd1 is located mainly on mitochondria (Supplementary Figs 16d and**
**17a, b). *Drosophila* Dosmit combines the features of Cisd1 and Cisd2**
**(Supplementary Fig. 16a)**

**Do the authors want to state that fly Dosmit is localized to mitochondria when**
**expressed in HeLa cells**

Reply: Indeed, we would like to indicate that mouse Cisd1 is located mainly in the
mitochondria, and that the fly homolog, Dosmit, is located in the mitochondria when
expressed in either fly S2 cells or in HeLa cells. This feature is similar to mouse
Cisd1.

**References**

- 1 Ruan, L. *et al.* Cytosolic proteostasis through importing of misfolded proteins
into mitochondria. *Nature* **543**, 443–446, doi:10.1038/nature21695 (2017).
- 2 Moehle, E. A., Shen, K. & Dillin, A. Mitochondrial proteostasis in the context
of cellular and organismal health and aging. *The Journal of biological*
*chemistry* **294**, 5396–5407, doi:10.1074/jbc.TM117.000893 (2019).
- 3 Matheoud, D. *et al.* Parkinson's Disease-Related Proteins PINK1 and Parkin
Repress Mitochondrial Antigen Presentation. *Cell* **166**, 314–327,
doi:10.1016/j.cell.2016.05.039 (2016).
- 4 Abuaita, B. H., Schultz, T. L. & O'Riordan, M. X. Mitochondria-Derived
Vesicles Deliver Antimicrobial Reactive Oxygen Species to Control

REVIEWERS' COMMENTS:

Reviewer #1 (Remarks to the Author):

The authors addressed my concerns adequately. They provide evidence that Dosmit is linked to mitochondrial proteostasis. Furthermore, they now carefully investigated the localization of Dosmit. The findings are interesting and raise a number of questions about the underlying molecular mechanisms for future studies.

Reviewer #2 (Remarks to the Author):

The authors have done a good job of addressing queries and criticisms. This is a robust and interesting study and I support publication. One final point to bring the authors' attention is to review the new Supplementary figure labels which related to Digitonin - currently, concentrations are marked um instead of uM.

Alex Whitworth

REVIEWERS' COMMENTS:

Reviewer #1 (Remarks to the Author):

The authors addressed my concerns adequately. They provide evidence that Dosmit is linked to mitochondrial proteostasis. Furthermore, they now carefully investigated the localization of Dosmit. The findings are interesting and raise a number of questions about the underlying molecular mechanisms for future studies.

Thank you for all of your suggestions and comments.

Reviewer #2 (Remarks to the Author):

The authors have done a good job of addressing queries and criticisms. This is a robust and interesting study and I support publication.

One final point to bring the authors' attention is to review the new Supplementary figure labels which related to Digitonin - currently, concentrations are marked um instead of μM .

Alex Whitworth

Thank you for your help reviewing this manuscript.

This error has now been corrected.